# A Bayesian Three-Cornered Hat (BTCH) Method: Improving the Terrestrial Evapotranspiration Estimation

**Xinlei He [1], Tongren Xu [1,*], Youlong Xia [2], Sayed M. Bateni [3], Zhixia Guo [1], Shaomin Liu [1], Kebiao Mao [4], Yuan Zhang [1], Huaize Feng [1] and Jingxue Zhao [1]**

1   State Key Laboratory of Earth Surface Processes and Resource Ecology, School of Natural Resources, Faculty of Geographical Science, Beijing Normal University, Beijing 100875, China; hxlbsd@mail.bnu.edu.cn (X.H.); 201621170058@mail.bnu.edu.cn (Z.G.); 201931051044@mail.bnu.edu.cn (S.L.); yuanzhang123@mail.bnu.edu.cn (Y.Z.); fenghuaize@mail.bnu.edu.cn (H.F.); 201821051096@mail.bnu.edu.cn (J.Z.)
2   I. M. Systems Group at Environmental Modeling Center (EMC), National Centers for Environmental Prediction (NCEP), College Park, MD 20741, USA; youlong.xia@noaa.gov
3   Department of Civil and Environmental Engineering and Water Resources Research Center, University of Hawaii at Manoa, Honolulu, HI 96822, USA; smbateni@hawaii.edu
4   National Hulunber Grassland Ecosystem Observation and Research Station, Institute of Agricultural Resources and Regional Planning, Chinese Academy of Agricultural Sciences, Beijing 100081, China; maokebiao@caas.cn
*   Correspondence: xutr@bnu.edu.cn; Tel.: +86-10-5880-7455 (ext. 1686)

**Abstract:** In this study, a Bayesian-based three-cornered hat (BTCH) method is developed to improve the estimation of terrestrial evapotranspiration (ET) by integrating multisource ET products without using any a priori knowledge. Ten long-term (30 years) gridded ET datasets from statistical or empirical, remotely-sensed, and land surface models over contiguous United States (CONUS) are integrated by the BTCH and ensemble mean (EM) methods. ET observations from eddy covariance towers ($ET_{EC}$) at AmeriFlux sites and ET values from the water balance method ($ET_{WB}$) are used to evaluate the BTCH- and EM-integrated ET estimates. Results indicate that BTCH performs better than EM and all the individual parent products. Moreover, the trend of BTCH-integrated ET estimates, and their influential factors (e.g., air temperature, normalized differential vegetation index, and precipitation) from 1982 to 2011 are analyzed by the Mann–Kendall method. Finally, the 30-year (1982 to 2011) total water storage anomaly (TWSA) in the Mississippi River Basin (MRB) is retrieved based on the BTCH-integrated ET estimates. The TWSA retrievals in this study agree well with those from the Gravity Recovery and Climate Experiment (GRACE).

**Keywords:** evapotranspiration; Bayesian-based three-cornered hat method; total water storage anomaly

## 1. Introduction

Evapotranspiration (ET) refers to the amount of water vapor evaporated from the land surface to the atmosphere [1,2]. The accurate estimation of ET is required for understanding the water resource management, water cycle, and climate change [3–6]. ET can be estimated directly by flux tower systems (e.g., FLUXNET, AmeriFlux, EuroFlux, HiWATER, etc.) [7–9]. However, these measurements have a sparse distribution and limited time periods. Hence, several approaches have been developed to estimate the spatial and temporal variability of ET over the regional and global scales. The existing

methods for estimating ET can be divided into three main groupings. The first group (called the statistical or empirical models) obtains regional ET based on flux tower observations [10–12]. The second group (called the remotely-sensed methods) estimates ET by incorporating remote sensing observations into the empirical models [13–16], surface energy balance equation [17,18], Penman–Monteith or Priestley–Taylor equation [19–21], and data assimilation methods [22–29]. The third group (named the land surface models) utilizes physical models (land surface models) or combines physical models with data assimilation algorithms to predict ET [30–33].

Using the abovementioned methods, various ET products (e.g., Gridded FLUXNET ET, GFET [10]; Moderate Resolution Imaging Spectroradiometer (MODIS) ET, MOD16 [34,35]; ETWatch [36,37]; Global Land Surface Satellite (GLASS) ET [38]; Global Land Evaporation and Amsterdam Model ET, GLEAM [39,40]; North American Land Data Assimilation System, NLDAS [31,41]) have been generated in the last few decades. However, there is a large difference among these ET products [13,42,43]. The uncertainties of ET products from the statistical or empirical, remotely sensed, and land surface models range from 4 to 15 mm/month [43].

Merging various ET products by the multimodel averaged method can generate an improved ET product with lower uncertainty [21,44]. Several studies have shown that the ET estimates from the multimodel averaged method are better than those of individual models [45–49]. Multimodel averaged ET datasets (e.g., LandFlux_EVAL) were compared with other ET products [14,38,50]. More sophisticated multimodel ensemble approaches, which consider the weights for independent algorithms based on in situ measurements have also been proposed to calculate ET. For example, Yao et al. (2014) [38] improved the ET estimates by merging five physical-based models within a Bayesian model averages (BMA) approach. Yao et al. (2017) [51] improved ET estimates by integrating semi-empirical ET algorithms with support vector machine (SVM). Hobeichi et al. (2018) [52] used the weighting approach to integrate the existing gridded ET products and produce a global ET product. However, the results of weighted fusion method are affected by the accuracy of in situ observations and the spatial scale mismatch between flux towers ET observations (with 100 meter spatial representativeness) and gridded ET products (usually produced with 0.25° resolution) [52,53].

The three-cornered hat (TCH) method has been used to assess the relative uncertainty of gridded datasets without any a priori knowledge [54]. The TCH approach considers cross correlation among ET products, and does not require the products to be independent [55]. Currently, the TCH approach has increasingly been used to evaluate uncertainties in soil moisture [19,56], total water storage [57], and ET [42,58] products over regional scales. The TCH approach has also been used in soil moisture integration applications, but it is rarely used in ET estimates [59].

In this study, a Bayesian-based three-cornered hat (BTCH) method is developed to integrate ET products without using any a priori knowledge. Ten long-term (30 years) gridded ET products from GFET, GLEAM, NLDAS-2, and NLDAS-testbed are integrated by the BTCH method over contiguous United States (CONUS). There are other ET products such as MOD16, ETWatch, and Simplified Surface Energy Balance (SSEBop) [35,60,61]. However, these ET products are not used in this study because they are available over shorter periods (i.e., less than 30 years). The BTCH-integrated ET product is compared with the ET observations from eddy covariance (EC) flux towers ($ET_{EC}$) and ET estimates from the water balance equation ($ET_{WB}$). Moreover, the 30-year (1982 to 2011) trends of ET product from BTCH and its influential climate factors (e.g., air temperature, normalized differential vegetation index NDVI, and precipitation) are studied by the Mann–Kendall method. Finally, using the BTCH-integrated ET, the 30-year total water storage anomaly (TWSA) in the Mississippi River Basin (MRB) is retrieved, and validated against the Gravity Recovery and Climate Experiment (GRACE) measurements.

The objectives of our study are to: (1) develop a more accurate ET product by integrating ten long-term gridded ET products within the BTCH method; (2) evaluate the BTCH-integrated ET estimates versus $ET_{EC}$ and $ET_{WB}$; (3) analyze the trend of 30-year BTCH-integrated ET product and its climate factors by the Mann–Kendall method; and (4) retrieve the 30-year total TWSA in the Mississippi River Basin (MRB) using the BTCH-integrated ET product.

## 2. Study Regions and Datasets

This study covers the contiguous United States (CONUS), ranging from 25.8° N to 50.8° N and 124.8° W to 67.8° W (Figure 1). The vegetation types of CONUS are forest, grassland, cropland, and shrubland. The land cover types are obtained from the MODIS land cover type product (MCD12Q1) (https://lpdaac.usgs.gov/products/mcd12q1v006/). The CONUS is composed of 12 National Weather Service River Forecast Centers (NWS RFCs): CB (Colorado), CN (California-Nevada), WG (West Gulf), MB (Missouri), AB (Arkansas), NC (North central), NW (Northwest), MA (Mid-Atlantic), SE (Southeast), NE (Northeast), LM (Lower Mississippi), and OH (Ohio). The integrated ET product is assessed using eddy covariance ET observations from 15 AmeriFlux sites. Characteristics of the 15 AmeriFlux sites are shown in Table 1. The energy closure imbalance of monthly ET data from 15 AmeriFlux sites was corrected by Jung et al. (2010) [10]. The locations of 12 RFCs and 15 AmeriFlux sites are shown in Figure 1.

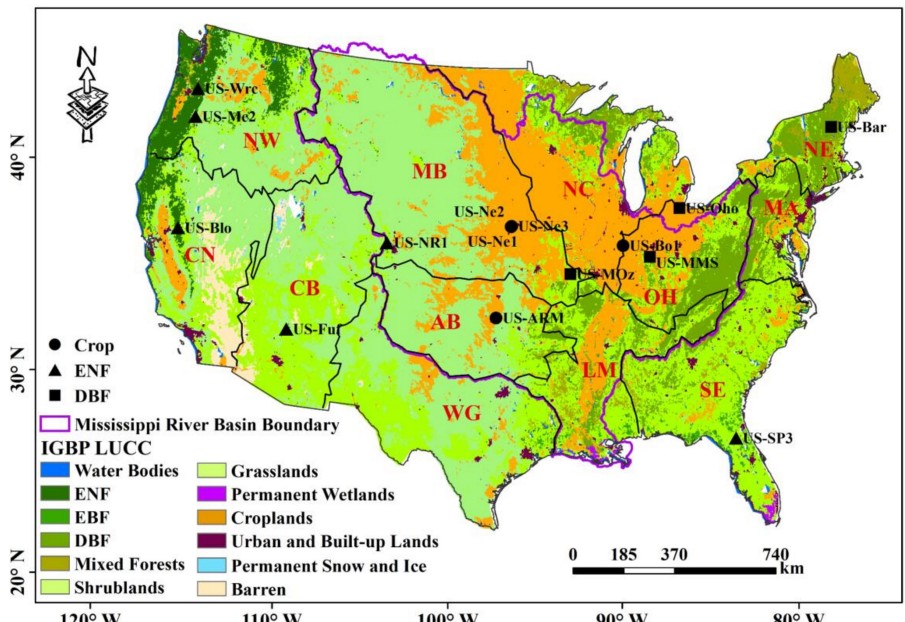

**Figure 1.** The land cover types and location of the 15 AmeriFlux stations over the contiguous United States (CONUS). DBF, EBF, and ENF represent deciduous broadleaf forest, evergreen broadleaf forest, and evergreen needleleaf forest, respectively.

**Table 1.** Characteristics of the 15 AmeriFlux stations over the contiguous United States.

| ID | Site Name | Period | Location (N°, W°) | Elevation (m) | Land Cover |
|---|---|---|---|---|---|
| US-Bar | Bartlett Experimental Forest | 2004–2005 | 44.06, -71.28 | 272 | DBF |
| US-MOz | Missouri Ozark | 2004–2006 | 38.74, -92.20 | 219 | DBF |
| US-MMS | Morgan Monroe State Forest | 2000–2004 | 39.32, -86.41 | 275 | DBF |
| US-Oho | Ohio Oak Openings | 2004–2005 | 41.55, -83.84 | 230 | DBF |
| US-Blo | Blodgett Forest | 2000–2006 | 38.89, -120.63 | 1315 | ENF |
| US-SP3 | Donaldson | 2000–2004 | 29.75, -82.16 | 50 | ENF |
| US-Fuf | Flagstaff Unmanaged Forest | 2005–2006 | 35.08, -111.76 | 2180 | ENF |
| US-Me2 | Metolius Pine | 2003–2005 | 44.45, -121.55 | 1253 | ENF |
| US-NR1 | Niwot Ridge | 2000–2003 | 40.03, -105.54 | 3050 | ENF |
| US-Wrc | Wind River | 2000–2006 | 45.82, -121.95 | 371 | ENF |
| US-ARM | ARM Southern Great Plains Main | 2003–2006 | 36.60, -97.48 | 314 | Cropland |
| US-Bo1 | Bondville | 2000–2006 | 40.00, -88.29 | 219 | Cropland |
| US-Ne1 | Mead Irrigated | 2001–2005 | 41.16, -96.47 | 361 | Cropland |
| US-Ne2 | Mead Irrigated Rotation | 2002–2005 | 41.16, -96.47 | 362 | Cropland |
| US-Ne3 | Mead Rainfed | 2001–2005 | 41.18, -96.44 | 363 | Cropland |

Ten ET products (namely, GFET, GLEAM, and four NLDAS-2 ET, and four NLDAS-testbed ET datasets) are integrated by the BTCH method (Table 2). GFET is generated by the Max Planck Institute for Biogeochemistry in Germany (https://www.bgc-jena.mpg.de/geodb/projects/Data.php). This product is derived from a data-driven model using global FLUXNET eddy covariance measurements. The GLEAM ET dataset is estimated from satellite-based data (i.e., soil moisture, land surface temperature, vegetation optical depth, and snow water equivalents) using the Priestley–Taylor equation over a global scale with spatial resolution of 0.25° [36,62,63]. The GLEAM ET products are available on the Global Land Evaporation Amsterdam Model archive (https://www.gleam.eu/).

The other eight ET products are sourced from physical models in NLDAS-2 and NLDAS-testbed. NLDAS-2 is composed of Noah28, Mosaic, VIC403, and SACs. NLDAS-tested contains Noah36, NoahMP36, VIC412, and CLSM25 [64–70]. Among all of these models, the Noah28, Noah36, NoahMP36, Mosaic, and CLSM25 models are developed within the soil vegetation atmosphere transfer (SVAT) scheme. VIC403, VIC412, and SAC are hydrological models [71–73]. All four models in NLDAS-2 or NLDAS-testbed used the same soil and vegetation types and vegetation cover fraction, but they were different in model parameter values [39]. NLDAS-testbed prepares an experimental system for the next generation of NLDAS, which is further advanced by upgrading the Noah and VIC models, and introducing a new model (i.e., CLSM25).

The auxiliary data used in this study include precipitation (P), air temperature (Ta), runoff (R), TWSA, soil moisture (SM), and normalized differential vegetation index (NDVI) data. The monthly gridded P and Ta datasets are obtained from National Centers for Environmental Information (NCEI) (https://www.ncei.noaa.gov) [74,75]. The regional monthly runoff data are downloaded from the United States Geological Survey (USGS) water watch platform (http://waterwatch.usgs.gov/). The daily NDVI data are derived from the Advanced Very High-Resolution Radiometer (AVHRR) instrument onboard the series of the National Oceanic and Atmospheric Administration (NOAA) satellites with the spatial resolution of 0.05° (https://climatedataguide.ucar.edu/climate-data/ndvi-normalized-difference-vegetation-index-noaa-avhrr). The AVHRR NDVI data were further averaged yearly using the time-average method to avoid cloud contaminations. The TWSA data are obtained from the GRACE Tellus website with the spatial resolution of 1° × 1° [76,77] (https://grace.jpl.nasa.gov/). The monthly soil moisture (SM, 1 meter) and ET values from CLM4 are also used as references to evaluate the integrated ET products. The related regional data are summarized in Table 2.

**Table 2.** Summary of regional datasets over the contiguous United States.

| Dataset | Data Source | Spatial, Temporal Resolution | Spatial, Temporal Coverage | References |
|---|---|---|---|---|
| P & Ta | NCEI | 4.16 km, monthly | CONUS, 1979–2012 | [74,75] |
| R | USGS | ~2°, daily | CONUS, 1982–2011 | [78] |
| TWSA | GRACE | ~1°, monthly | Global, 2003–2016 | [76] |
| SM & ET | CLM4 | 12.5 km, monthly | CONUS, 1980–2014 | [79] |
| NDVI | AVHRR | 0.05°, daily | Global, 1981–2018 | [80] |
| SE ET | GFET | 50 km, monthly | Global, 1982–2011 | [10,81] |
| RSM ET | GLEAM | 25 km, daily | Global, 1982–2017 | [37] |
| NLDAS-2 ET | Noah28 SAC Mosaic VIC403 | 12.5 km, monthly | CONUS, 1979–2013 | [69,70] |
| NLDAS-testbed ET | CLSM25 Noah36 NoahMP36 VIC412 | 12.5 km, monthly | CONUS, 1979–2013 | [79] |

SE, statistical or empirical models; RSM, remotely-sensed model; and NLDAS, North American Land Data Assimilation System.

## 3. Methodology

### 3.1. Bayesian-Based Three-Cornered Hat (BTCH) Method

In this study, the BTCH method is utilized to generate an accurate ET product by combining various ET datasets [82,83]. The probability density function (PDF) for the *i*th ET product ($ET_i$) can be expressed as,

$$p(ET_i|ET_t) = \frac{1}{\sigma_i \sqrt{2\pi}} \exp\left[-\frac{\varepsilon_i^2}{2\sigma_i^2}\right] = L(ET_t|ET_i) \ (\varepsilon_i = ET_i - ET_t), \tag{1}$$

where $ET_t$ is the true value of ET. $\varepsilon_i$ and $\sigma_i$ are a zero-mean white noise and error variance of the *i*th ET product, respectively. $L(\bullet)$ is the likelihood function.

Similarly, the PDF for *j*th ET product ($ET_j$) can be expressed as,

$$p(ET_j|ET_t) = \frac{1}{\sigma_j \sqrt{2\pi}} \exp\left[-\frac{\varepsilon_j^2}{2\sigma_j^2}\right] = L(ET_t|ET_j) \ (\varepsilon_j = ET_j - ET_t), \tag{2}$$

where $\varepsilon_j$ and $\sigma_j$ are a zero-mean white noise and error variance of the *j*th ET product.

The maximum likelihood of true ET ($ET_t$) is the maximum value of its joint probability distribution,

$$\max L(ET_t|ET_i, ET_j) = p(ET_i|ET_t)p(ET_j|ET_t) = \frac{1}{2\pi\sigma_i\sigma_j} \exp\left[-\frac{\varepsilon_i^2}{2\sigma_i^2} - \frac{\varepsilon_j^2}{2\sigma_j^2}\right], \tag{3}$$

In order to obtain the maximum likelihood value of $ET_t$ in Equation (3), the cost function *J* is defined as,

$$J(ETt) = \frac{\varepsilon_i^2}{2\sigma_i^2} + \frac{\varepsilon_j^2}{2\sigma_j^2} = \frac{1}{2}\left[\frac{(ETi - ETt)^2}{\sigma_i^2} + \frac{(ETj - ETt)^2}{\sigma_j^2}\right], \tag{4}$$

By setting the first variation of $J(ET_t)$ to zero (i.e., $\delta J(ET_t) = 0$), $ET_t$ can be obtained as,

$$ETt = \frac{\sigma_i^2}{\sigma_i^2 + \sigma_j^2}ETi + \frac{\sigma_j^2}{\sigma_i^2 + \sigma_j^2}ETj, \tag{5}$$

If we define $ET_t = w_iET_i + w_jET_j$, then

$$wi = \frac{\sigma_i^2}{\sigma_i^2 + \sigma_j^2}, wj = \frac{\sigma_j^2}{\sigma_i^2 + \sigma_j^2}, \tag{6}$$

Similar to Equation (5), $ET_t$ can be obtained via $ET_t = w_1ET_1 + \cdots + w_NET_N$ when there are *N* sets of ET products. The weight of each ET product (e.g., $w_k$) can be obtained by minimizing a similar cost function (Equation (4)) as,

$$w_k = \frac{\prod\limits_{i=1,i\neq k}^{N} \sigma_i^2}{\sum\limits_{k=1}^{N}\left(\prod\limits_{i=1,i\neq k}^{N} \sigma_i^2\right)}, \tag{7}$$

The error covariance ($\sigma_i$) of each ET product can be obtained using the three-cornered hat (TCH) method.

Since $ET_t$ is not available, the difference between (*N*-1) ET products and a reference ET product ($ET_R$) (chosen arbitrarily from *N* ET products) can be expressed as,

$$\mathbf{Y}_{i,M} = ETi - ET_R = \varepsilon_i - \varepsilon_R \cdot i = 1, 2, \ldots, N-1, \tag{8}$$

where **Y** is the difference matrix with (*N*-1) rows and *M* columns (*M* is the total number of time samples in each ET product). The covariance matrix (**S**) of **Y** is defined as [84],

$$\mathbf{S} = cov(\mathbf{Y}), \tag{9}$$

where $cov(\bullet)$ is the covariance operator. The unknown covariance matrix of the individual noise **R** is related to **S** via,

$$\mathbf{S} = \mathbf{J} \cdot \mathbf{R} \cdot \mathbf{J}^T, \tag{10}$$

where **J** is the identity matrix, and can be defined as,

$$\mathbf{J}_{N-1,N} = \begin{bmatrix} 1 & 0 & \cdots & 0 & -1 \\ 0 & 1 & \cdots & 0 & -1 \\ \vdots & \vdots & \vdots & \vdots & \vdots \\ 0 & 0 & 0 & \cdots & -1 \end{bmatrix}, \tag{11}$$

and matrix **R** is defined as,

$$\mathbf{R} = \begin{bmatrix} \sigma_{11} & \sigma_{12} & \cdots & \sigma_{1N} \\ \sigma_{12} & \sigma_{22} & \cdots & \sigma_{2N} \\ \vdots & \vdots & \vdots & \vdots \\ \sigma_{1N} & \sigma_{2N} & \cdots & \sigma_{NN} \end{bmatrix}, \tag{12}$$

where $\sigma_{ij} = cov(\varepsilon_i, \varepsilon_j)$. There are $N \times (N + 1)/2$ unknowns in Equation (10) (number of distinct elements of **R**), but there are only $N \times (N - 1)/2$ equations (number of distinct elements of **S**). Thus, there remain $N$ "free" parameters that must be reasonably determined to obtain a unique solution [84]. To determine the $N$ free parameters, the following objective function is minimized based on the Kuhn–Tucker theorem.

The Kuhn–Tucker theorem is proposed by Galindo and Palacio (1999) [85], and the objective function is defined as,

$$F_{(\sigma_{1N}, \cdots, \sigma_{NN})} = \frac{1}{\mathbf{K}^2} \cdot \sum_{i<j}^{N} (\sigma_{ij})^2, \tag{13}$$

where $\mathbf{K} = \sqrt[N-1]{\det(\mathbf{S})}$. The constraint function is given by,

$$H_{(\sigma_{1N}, \dots, \sigma_{NN})} = -\frac{|\mathbf{Q}|}{|\mathbf{S}| \cdot \mathbf{K}} < 0, \tag{14}$$

where **Q** is a diagonal matrix with elements $\sigma_{11}, \cdots, \sigma_{NN}$ on its diagonal. These arrays can be calculated by minimizing Equation (13) through the initial condition iterations [86]. The square root of the diagonal elements of **R** (i.e., $\sigma_{11}, \sigma_{22}, \dots, \sigma_{NN}$) represent the relative uncertainty of each ET product [43]. The readers are referred to Long et al. (2014) [42] and Xu et al. (2019) [43] for detailed information on the TCH approach.

The ensemble mean (EM) method is used to also integrate the ten long-term gridded ET products and generate an ET product. This method just simply calculates the average of ET products as follows,

$$ET_{\mathrm{EM}} = \frac{1}{N} \sum_{i=1}^{N} ETi, \tag{15}$$

### 3.2. Water Balance Budget Method

The water budget equation can be expressed as,

$$ET_{\mathrm{WB}}(t) = P(t) - R(t) - \mathrm{TWSC}(t), \tag{16}$$

where $t$ is the time step (month), $ET_{WB}$ is the ET (mm/month) estimates from the water budget equation, TWSC is the total water storage change (mm/month), and $P$ and $R$ are the precipitation (mm/month) and runoff (mm/month), respectively. According to Zhang et al. (2010) [87] and Velpuri et al. (2013) [88], $ET_{WB}$ values can be used as the reference data to validate ET estimates at multiyear scale.

*3.3. Long-Term Total Water Storage Anomaly Reconstruction*

The regional TWSC data can be expressed as follows,

$$TWSC(t) = \frac{TWSA(t+1) - TWSA(t-1)}{2\Delta t}, \tag{17}$$

where TWSA is the total water storage anomaly and, on the one hand, can be obtained from GRACE data. On the other hand, TWSC can be calculated from Equation (16) using $P$, $R$, and ET, and TWSA can be calculated via,

$$TWSA(t) = \frac{TWSC(1) + 2TWSC(2) + \cdots + tTWSC(t)}{t+1}, \tag{18}$$

As shown in Equation (18), the uncertainty of TWSA estimates is due to cumulative errors in TWSC estimates, and TWSC uncertainties can be attributed to errors in $P$, $R$, and ET. Thus, the *PER* (precipitation, evapotranspiration, and runoff) method proposed by Zeng, (1999) [89] is applied to Equation (18) to eliminate time accumulative errors, especially systematic bias,

$$TWSC(t) = P(t) - R(t) - ET_{WB}(t) - (\overline{P(t)} - \overline{ET(t)}) + \overline{R(t)}, \tag{19}$$

where $\overline{P}$, $\overline{ET}$, and $\overline{R}$ represents mean values of $P$, ET, and $R$, respectively.

The Mann–Kendall method is used to analysis linear trend of the BTCH-integrated ET products [90, 91]. The Mann–Kendall test is a non-parametric method to identify trends of time series and is widely used in previous studies [92–95].

The standard deviation (SD), root mean square deviation (RMSE), and correlation coefficients (R) metrics are used to access the performances of VDA approach,

$$SD = \sqrt{\frac{1}{N} \sum_{t=1}^{N} (P_t - \overline{O})^2}, \tag{20}$$

$$RMSD = \sqrt{\frac{1}{N} \sum_{t=1}^{N} (P_t - O_t)^2}, \tag{21}$$

$$R = \frac{\sum_{t=1}^{N} (Pt - \overline{P})(Ot - \overline{O})}{\sqrt{\sum_{t=1}^{N} (Pt - \overline{P})^2} \sqrt{\sum_{t=1}^{N} (Ot - \overline{O})^2}}, \tag{22}$$

where $P_t$ and $O_t$ are the predicted and observed values at time step $t$, respectively.

Partial correlation analysis [96,97] is used to obtain the correlation between collinear hydrologic variables (herein, Ta, $P$, and NDVI) that affect ET. The partial correlation analysis eliminates collinearity among the hydrologic variables (i.e., Ta, $P$, and NDVI) controlling ET. This analysis allows the impact of one or more variables (e.g., Ta and NDVI) to be eliminated when examining the relationship between another pair of variables (e.g., ET and $P$) [98,99]. Hence, the partial correlation coefficient quantifies the correlation between two variables (e.g., ET and $P$), controlling the other variables (i.e., Ta and NDVI).

The partial correlation between the variables $x$ and $y$, which are conditioned on the variable $z$ can be calculated by,

$$Rxy.z = \frac{Rxy - RxzRyz}{\sqrt{(1 - R_{xz}^2)(1 - R_{yz}^2)}},$$

(23)

where $R_{xy,z}$ is the partial correlation between $x$ and $y$, given the control $z$. The variables $R_{xy}$, $R_{xz}$, $R_{yz}$ are the correlation between $x$ and $y$, $x$ and $z$, and $y$ and $z$, respectively. More details on the formulation of the partial correlation method is provided by [96] and [97].

## 4. Results and Discussions

### 4.1. ET Product from the BTCH Method

Figure 2 shows the 30-year (1982–2011) averaged ET estimates from the BTCH method over the CONUS. The spatial patterns of retrieved ET are fairly consistent with those of rainfall and to a lesser extent vegetation density. The ET estimates in the southeast of CONUS (wet or densely vegetated areas) are relatively larger than those in the center and north of CONUS (dry or sparsely vegetated areas). Figure 3 shows the 30-year averaged NDVI and precipitation over the CONUS. Higher ET estimates in the southeast of CONUS are mainly due to heavy precipitation and dense vegetation cover (Figure 3). Figure 2 indicates that ET values of 100 to 400 mm/year cover about 60% of the areas of CONUS. The extreme large ET values (>700 mm/year) can only be found in the southeast of CONUS.

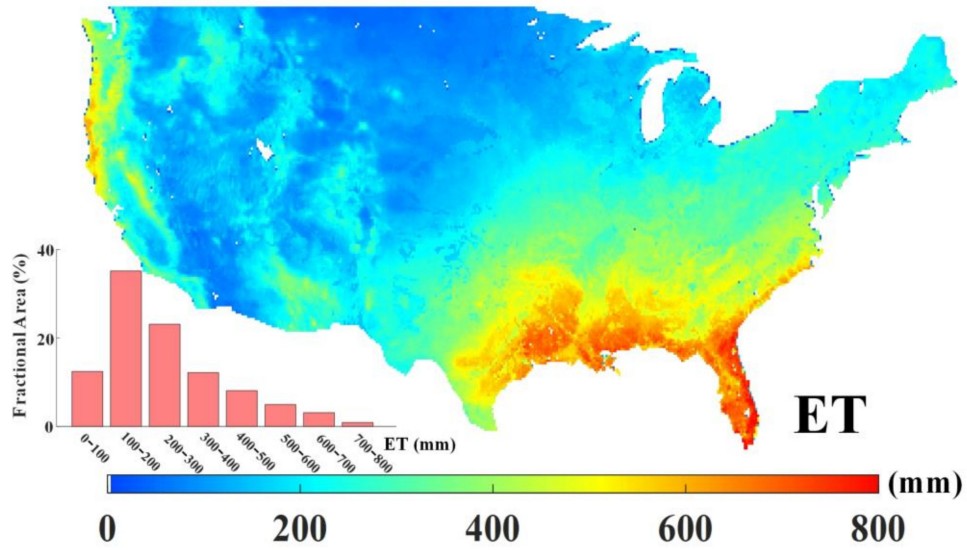

**Figure 2.** Spatial distribution of 30-year averaged evapotranspiration (ET) from the Bayesian-based three-cornered hat (BTCH) method.

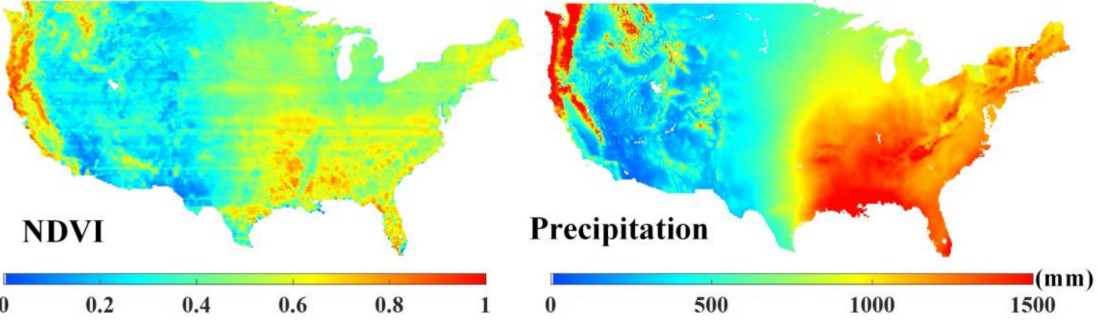

**Figure 3.** Maps of 30-year averaged normalized differential vegetation index (NDVI) and precipitation over the contiguous United States.

Figure 4 shows the statistics of BTCH-integrated, EM, and ten parent ET products as compared with flux tower measurements (ET$_{EC}$). As indicated, ET estimates from the BTCH method are closer to the observations as compared with other ET products. Since GFET is upscaled based on FLUXNET tower observations, it is also close to the AmeriFlux ET observations. The ET estimates from BTCH have higher R and lower RMSD as compared with those of EM and individual algorithms over the three land cover types. For DBF, ENF, and Crop, the RMSDs of ET estimates from BTCH are respectively 11.25, 15.13, and 17.25 mm/month, which are 31.28%, 20.16%, and 22.05% lower than those of 16.37, 18.95, and 22.13 mm/month from EM.

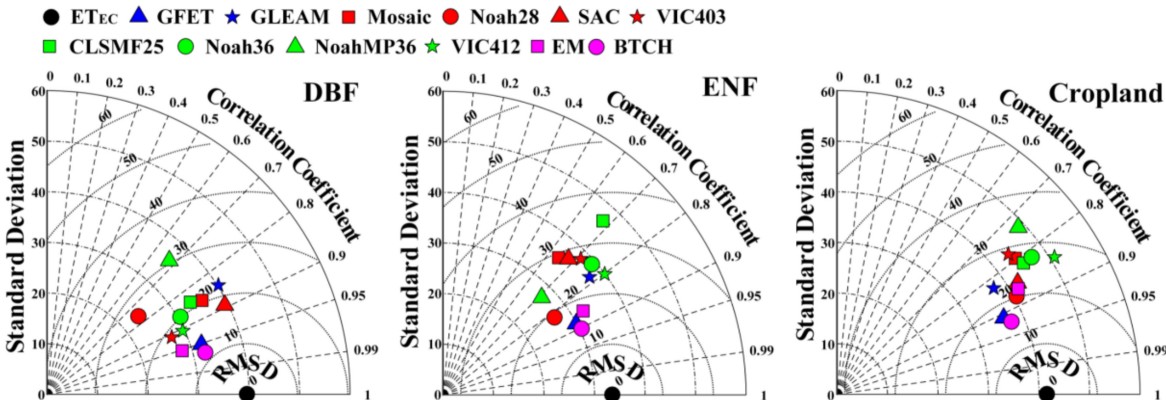

**Figure 4.** Comparison of monthly BTCH-integrated, ensemble mean (EM), and ten parent evapotranspiration (ET) datasets with flux tower observations. There are 144, 336, and 300 samples for the deciduous broadleaf forest (DBF), evergreen broadleaf forest (ENF), and cropland, respectively.

ET$_{WB}$ values are also used to assess the accuracy of ET estimates from BTCH, EM, and ten individual models at the basin scale in Figure 5. As indicated, BTCH performs better than EM and other models. ET products from BTCH and Noah28 are closest to the ET$_{WB}$ estimates, followed by VIC412 and GFET. For the 30-year comparisons (1982 to 2011), the average RMSD of ET product from the BTCH method is 24.64 mm/year, which is 53.46% lower than the RMSD of 52.94 mm/year from the EM method.

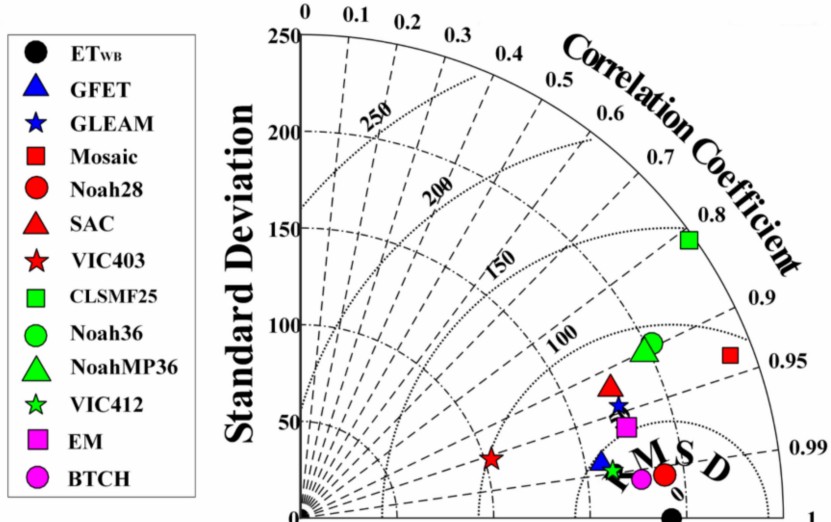

**Figure 5.** Comparison of annual BTCH-integrated, ensemble mean (EM), and ten parent evapotranspiration (ET) dataset with ET from water balance method from 1982 to 2011. There are 360 samples over the contiguous United States.

The weight (%) of each ET dataset has a key role in ET integration by the BTCH method. The weights of ten ET products over the main four land covers (cropland, grassland, ENF, and DBF) are shown in Figure 6. As indicated, GFET (22%), Noah28 (15%), and GLEAM (14%) have the largest weight over all land covers in the whole year. The relative contributions vary for different land cover types. For example, Noah28 has the highest weight (24%) over cropland; GFET has the highest weight (27%) over grassland; GFET and NoahMP36 have the highest weight (16%) over ENF; and GFET has the highest weight (24%) over DBF. GFET has the highest weight in different seasons. For different LSMs, Noah28 and Noah36 have the largest weight of 15% and 10% in a whole year, while VIC403 has lowest weight (5%). VIC412 increased weight from 5% to 8% as compared with VIC403 in whole year.

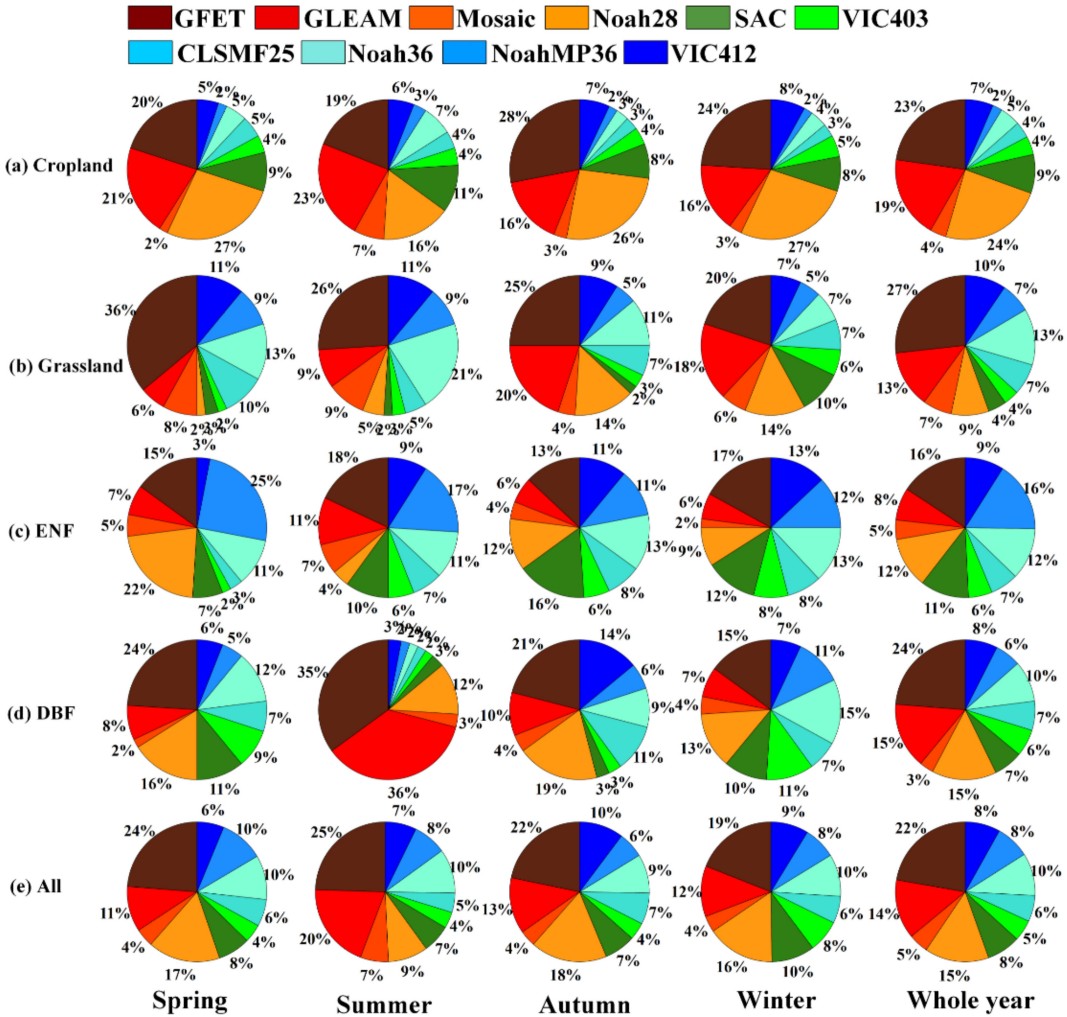

**Figure 6.** Weight (%) of each evapotranspiration (ET) product in the Bayesian-based three-cornered hat (BTCH) method.

## 4.2. Relationships between BTCH-Integrated ET and Climate Factors

The Mann–Kendall trend test was used to investigate the temporal variations of BTCH-integrated ET along with dominant climate factors (i.e., air temperature, NDVI, and precipitation) in the CONUS from 1982 to 2011(Figure 7). The black points represent 95% level of significance. The positive and negative values on the maps indicate increasing and decreasing trends, respectively. As shown, air temperature increased about 0.06 °C/year over the southwest CONUS, and decreased ~0.03 °C/year over the northwest CONUS. NDVI shows a positive trend over the southeast and west coast, whereas a negative trend was found in southwest areas. The precipitation increased (decreased) ~0.12 mm/year over the northeast (southeast) CONUS. As shown, there is a maximum of 0.04 mm/year growth

(reduction) in the BTCH-integrated ET over center, northeast, and west coast (southwest) areas. The growing trend of ET could be caused by increased vegetation density (NDVI) and precipitation. The decreased air temperature in the north and west coast areas could be caused by the "cooling effect" of increased ET. In the southwest areas (e.g., Texas), ET, NDVI, and precipitation are all decreased, indicating the continuous drought in this area. In contrast, Ta is increased in the southwest area (e.g., Texas). Because the land surface is dry, the available energy at the surface is mostly dissipated through sensible heat flux, which heats the atmosphere.

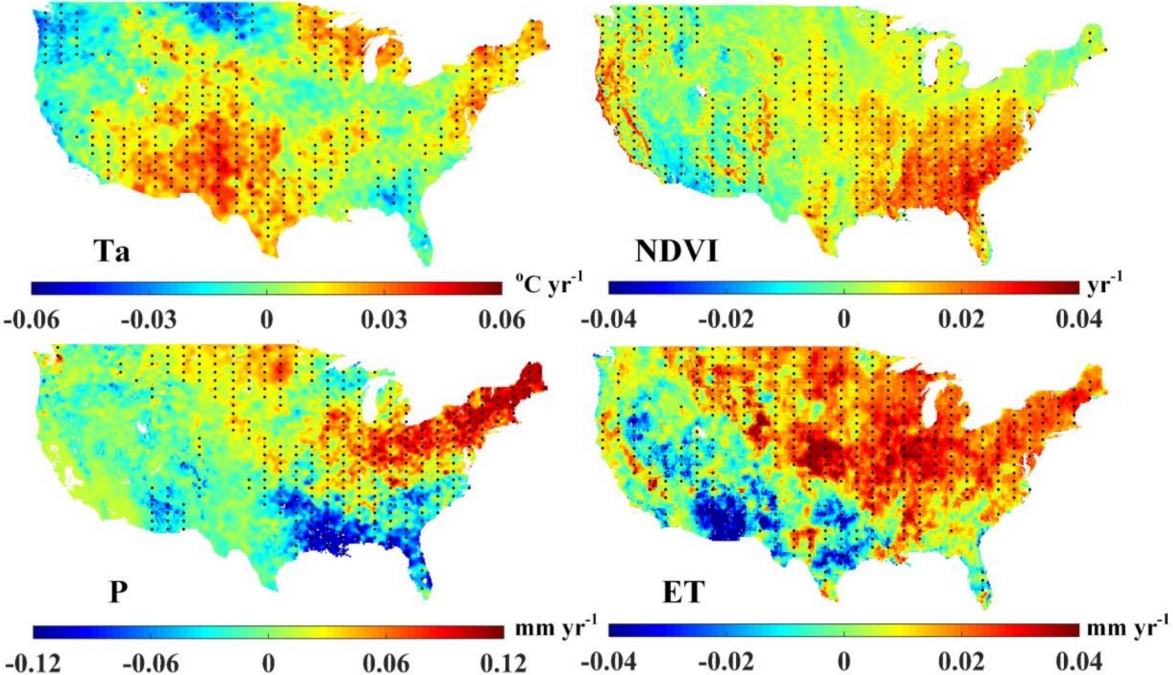

**Figure 7.** The spatial pattern of trends in air temperature (Ta), normalized differential vegetation index (NDVI), precipitation (*P*), and evapotranspiration (ET) over the contiguous United States from 1982 to 2011. The black points represent 95% level of significance.

The results also show different trends of ET among the RFCs. Over Colorado and California-Nevada, the ET estimates show a small decrease over the 30-year period (1982 to 2011). While, there is a small increase in the ET estimates over Missouri, North Central, Mid-Atlantic, Northeast, Lower Mississippi, and Ohio over the same period. Texas is mainly covered by the West Gulf and Arkansas basins, and experienced the most extreme one-year drought on record in 2011. The extreme one-year drought in 2011 has a large impact on the ET estimates [100,101]. Milly and Dunne (2001) [102] showed that increased ET in Mississippi has both climatological and anthropogenic dimensions. Walter et al. (2004) [103] studied ET across the CONUS based on precipitation and stream discharge data and showed that the ET has increased over the past 50 years. This study also reported an increasing trend in ET over the Mississippi basin and CONUS from 1982 to 2011, which is consistent with those of [102] and [103].

Plots of 30-year-averaged ET against Ta, *P*, and NDVI over the 12 RFCs are shown in Figure 8. As shown, in general, ET grows by an increase in Ta, *P*, and NDVI over the 12 RFCs. ET estimates over Southeast, Lower Mississippi, and Ohio are higher than those of Colorado and California-Nevada because of higher vegetation density, precipitation, and air temperature. In the West Gulf, low vegetation density and precipitation limit ET in spite of high air temperature. These results are consistent with those of [23] and [104].

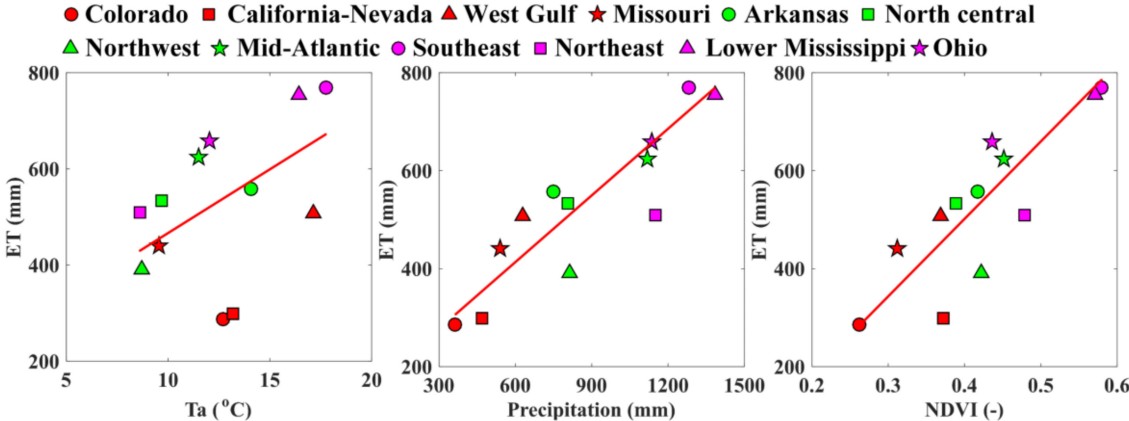

**Figure 8.** Plots of 30-year-averaged evapotranspiration (ET) against air temperature (Ta), precipitation (*P*), and normalized differential vegetation index (NDVI) over the 12 river forecast centers (RFCs). The red line represents the fitted linear regression.

Figure 9 shows the partial correlation coefficients of the annual ET estimates from BTCH with NDVI, precipitation, and air temperature from 1982 to 2011. As indicated, the annual ET estimates is positively correlated with precipitation and NDVI in the southwest CONUS, implying that the rise in these variables increase ET, and vice versa. This is because at dry or slightly vegetated conditions (e.g., southwest areas, Figure 3), ET is water limited and is mainly controlled by the surface state variables (i.e., soil moisture and NDVI).

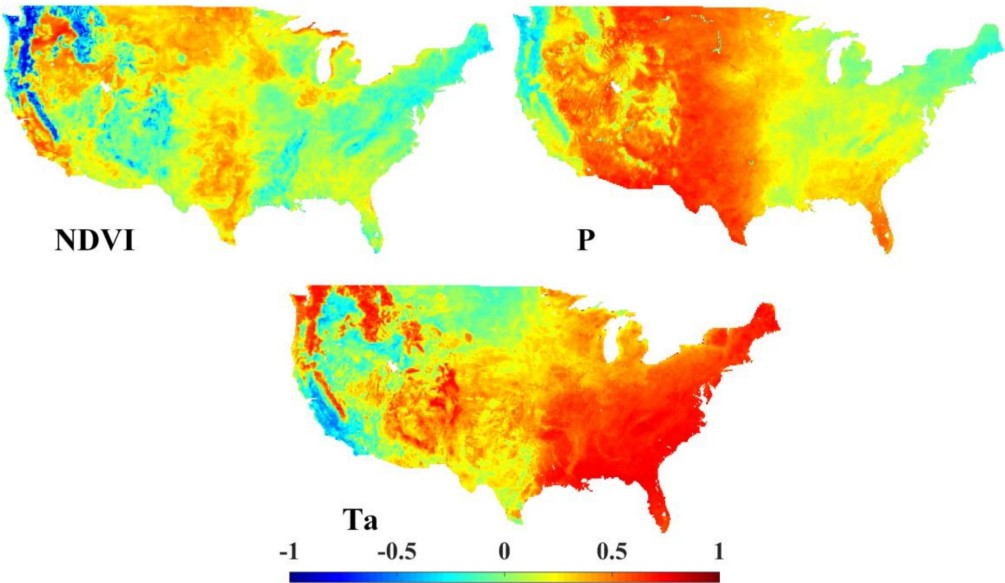

**Figure 9.** Partial correlation coefficients (R) between the annual evapotranspiration (ET) and air temperature (Ta), precipitation (*P*), and normalized differential vegetation index (NDVI).

The ET estimates are positively correlated with air temperature in the northwest, east, and southeast CONUS (wet or densely vegetated conditions, Figure 3). This happens because in wet or densely planted conditions (e.g., northwest, east, and southeast areas), ET is energy limited and is mainly influenced by atmospheric state variables (i.e., air temperature and specific humidity) [105,106].

*4.3. Interannual Variations of BTCH-Integrated ET Estimates*

Figure 10 shows time series of the annual BTCH-integrated ET estimates, soil moisture from CLM4, and precipitation anomalies over the twelve RFCs for 1982 to 2011. As indicated, ET anomalies

are consistent with those of soil moisture and precipitation. ET anomalies rise with the increase of soil moisture and precipitation anomalies, and vice versa. Over relatively dry RFCs (i.e., Colorado and California-Nevada), ET increases rapidly with precipitation (e.g., 1983, 1998, and 2005). While, there is a lower consistency between ET and soil moisture over relatively wet RFCs (i.e., Northeast, Lower Mississippi). This is because over relatively dry RFCs, ET is water limited and is mainly controlled by the land surface state variable (i.e., soil moisture). Therefore, the coupling between ET and soil moisture (precipitation) is more robust. In contrast, over relatively wet RFCs, ET is energy limited and is mainly affected by the atmospheric state variables (i.e., air temperature and specific humidity). Hence, the coupling between ET and soil moisture (precipitation) becomes weak.

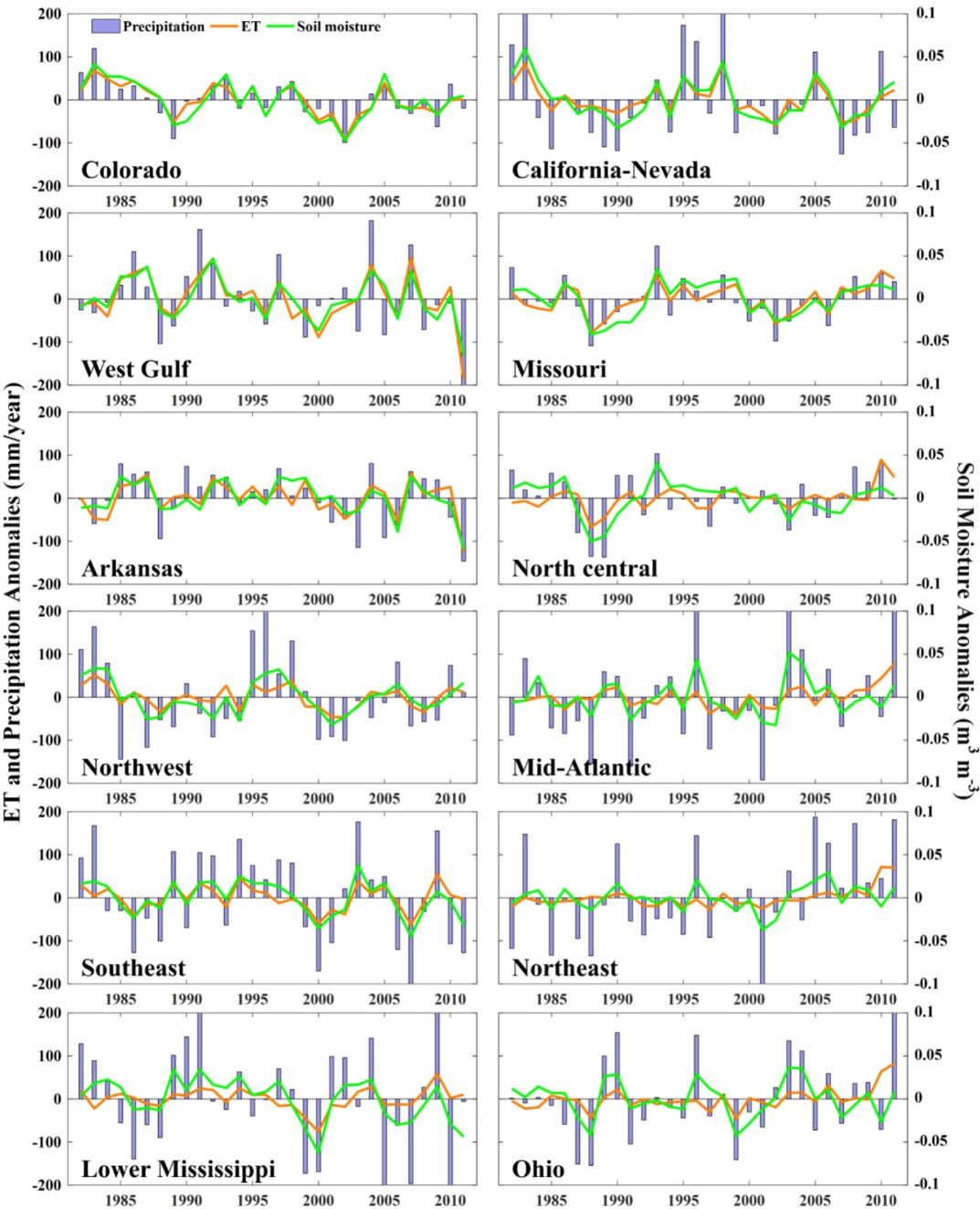

**Figure 10.** Annual variations of BTCH-integrated evapotranspiration (ET), soil moisture, and precipitation anomalies for 12 river forecast centers (RFCs) from 1982 to 2011.

### 4.4. Long-Term Reconstruction of GRACE Total Water Storage Anomaly

Using BTCH-integrated ET, precipitation, and runoff datasets, the TWSA over the Mississippi River Basin (MRB) is estimated from Equation (18). The long-term (1982–2011) TWSA estimates from the *PER* method are shown in Figure 11 (top). In general, the TWS estimates increased by 0.06 mm/year from 1982 to 2011. TWS has a decreasing trend prior to 1993 as represented by the negative TWSA estimates. In contrast, TWS increases after 1993 as reflected in the positive TWSA retrievals.

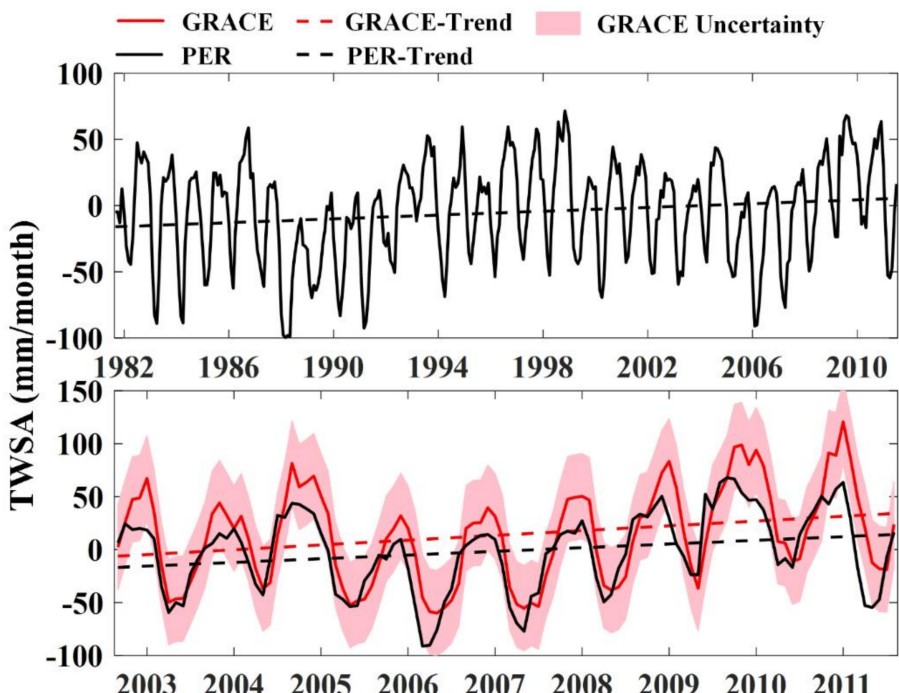

**Figure 11.** Time series of the total water storage anomaly (TWSA) retrievals over the Mississippi River Basin (MRB) from 1982 to 2011 (top). Comparison of the TWSA estimates with those of the Gravity Recovery and Climate Experiment (GRACE) from 2003 to 2011 (bottom).

The retrieved TWSA is compared with the GRACE-derived TWSA from 2002 to 2011 over the MRB. As shown in Figure 11 (bottom), the TWSA retrievals agree well with those of GRACE in terms of magnitude, as well as seasonal and long-term variations. The RMSDs and R of TWSA estimates from *PER* method are 32.92 mm/month and 0.82. From 2002 to 2011, on average, *PER* TWS estimates increased by ~0.32 (mm/year), which is close to the TWS increase of ~0.38 (mm/year) from GRACE. The misfit between the GRACE and *PER* TWSA retrievals is due to uncertainties in the water budget balance components (i.e., ET, precipitation, and runoff) and GRACE data.

### 4.5. Time Window of BTCH Method

As indicated by Xu et al. (2019) [43], uncertainties of ET estimates from 12 ET products (including one, three, and eight products from the machine learning model, remotely sensed observations, and land surface models, respectively) change with seasons. It is evident that the seasonal variations in the uncertainty of ET estimates from different methods affect ET integration by the BTCH method. Herein, the impact of time window on the accuracy of BTCH-integrated ET estimates is evaluated. The chosen time windows are 1, 2, 3, 6, and 12 months (i.e., ET integration by BTCH every 1, 2, 3, 6, and 12 months). Two-month time windows are December to January, February to March, . . . , and October to November. Three-month time windows are spring (April to June), summer (July to September), autumn (October to December), and winter (January to March). Six-month time windows are growth (May to October) and non-growth (November to April) seasons.

Figure 12 shows the monthly ET estimates from BTCH for the six tested time windows over DBF, ENF, and cropland. As can be seen, the integrated ET products for different time windows are comparable in magnitude and temporal variation. The ET estimates increase with the vegetation growth and reach their maximum in mid–July. Then, ET reduces as the vegetation decays. As indicated, the results of BTCH method with time window are improved significantly as compared with the EM methods. ET estimates from BTCH with one-month and two-month time windows are closer to the observations as compared with those with three-month and six-month time windows. Over ENF, ET estimates are remarkably lower than observations during the whole year. For cropland and DBF, ET values are significantly underestimated during the second half of the year. This is because the Noah28, VIC403, NoahMP36, and VIC412 models typically underestimate ET and fail to capture its seasonal dynamics over DBF and ENF [43]. Overall, the BTCH method with the one-month time window provides the most accurate ET estimates. This is because the one-month time window is able to capture seasonal variations in uncertainties of ET estimates better than those of two-month, three-month, six-month, and 12-month time windows, and EM.

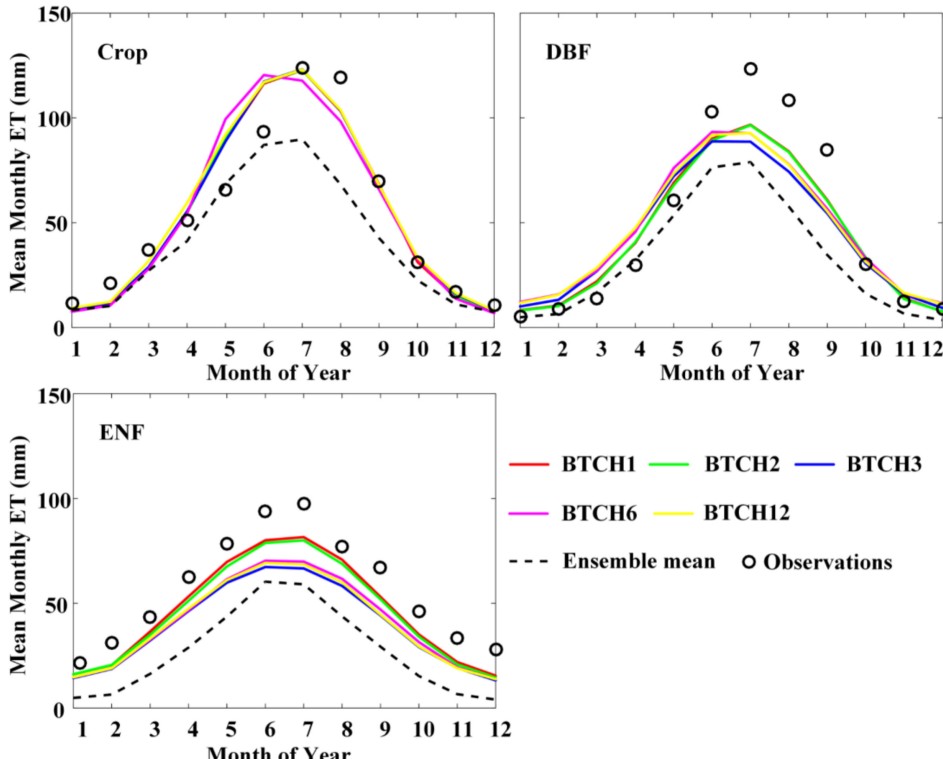

**Figure 12.** Monthly variations of evapotranspiration (ET) estimates from Bayesian-based three-cornered hat (BTCH) for different time windows, and ensemble mean (EM). BTCH1, BTCH2, BTCH3, BTCH6, and BTCH12 denote 1-month, 2-month, 3-month, 6-month, and 12-month time windows, respectively. The black circles are observations from eddy covariance flux towers.

Figure 13 compares statistical metrics of BTCH-integrated ET values from 1982 to 2011 for different time windows. The ET derived from water balance method ($ET_{WB}$) and ET from CLM4 ($ET_{CLM4}$) are used as the reference data to quantify different integration results. The results of EM are also shown for comparison. As indicated, the BTCH-integrated ET estimates are closer to observation, implying that BTCH outperforms EM. ET estimates from BTCH1 have the lowest RMSD and standard deviation, followed by those of BTCH2, BTCH3, BTCH6, and BTCH12. The seasonal variations in uncertainties of ET estimates can be efficiently used by the BTCH method to reduce the uncertainties in the integrated ET product.

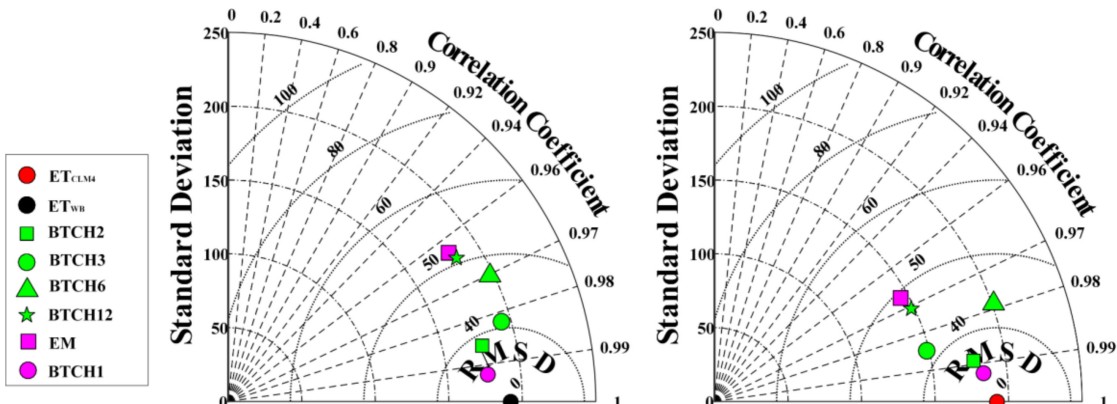

**Figure 13.** Comparison of evapotranspiration (ET) estimates (1982 to 2011) from the ensemble mean (EM) and Bayesian-based three-cornered hat (BTCH) methods for different time windows.

Figure 14 shows uncertainty of ET retrievals from BTCH over the CONUS, for the five different time windows. For comparison, uncertainty of ET estimates from EM is also shown in Figure 14. The TCH method is used to calculate the uncertainties of ET estimates from BTCH and EM. As indicated, the uncertainties of ET retrievals from BTCH1 to BTCH12 are lower than those of EM. The ET product from BTCH1 has the lowest uncertainty, followed by BTCH2, BTCH3, BTCH6, and BTCH12. The individual ET products have lower uncertainties in the northwest as compared with the southeast and center of CONUS [43]. However, BTCH1 can take advantage of seasonal ET uncertainties and eliminate high uncertainties of the other BTCH and EM products.

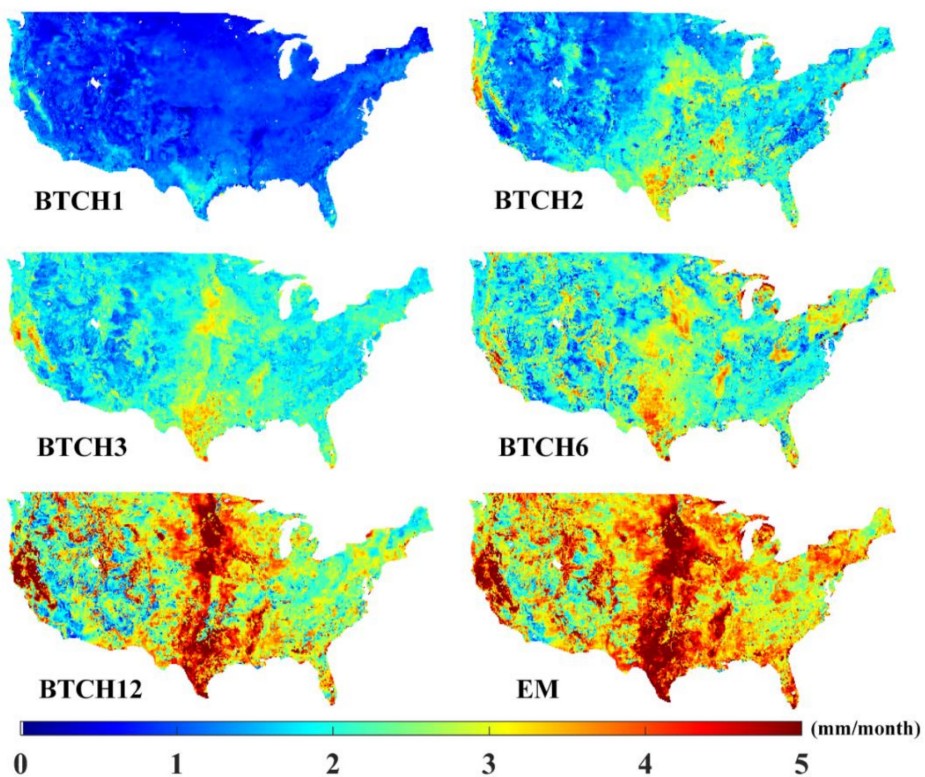

**Figure 14.** The uncertainties of evapotranspiration (ET) products from the ensemble mean (EM) and Bayesian-based three-cornered hat (BTCH) methods with different time windows over the contiguous United States.

## 5. Conclusions

In this study, a Bayesian-based three-cornered hat (BTCH) method is used to integrate ten long-term (30 years) gridded ET datasets (i.e., GFET, GLEAM, eight physical model outputs from NLDAS-2, and NLDAS-tested) without using any a priori knowledge.

The performance of the BTCH method is evaluated using ET measurements from AmeriFlux eddy covariance ($ET_{EC}$) stations and ET derived from the water balance method ($ET_{WB}$). The results show that BTCH is able to capture the seasonal variations of ET estimates more accurately than the ensemble mean (EM) method. The BTCH-integrated ET estimates have the RMSD of 14.54 mm/month, which is 24.07% lower than that of the RMSD of 19.15 mm/month from EM.

BTCH-integrated ET shows a maximum increase of ~0.04 (mm/year) over the center, northeast, and west coast of CONUS and a maximum decrease of ~0.04 (mm/year) over southwest areas. The positive trend of ET in the Mississippi River Basin from 1982 to 2011 is caused by increased vegetation density (NDVI) and precipitation. The decreased ET in the southwest areas is due to the reduction in NDVI and precipitation. The annual BTCH-integrated ET is positively correlated with precipitation and NDVI in dry or sparse vegetated conditions, while it is positively correlated with air temperature wet or densely vegetated conditions. This is because at dry or slightly vegetated conditions, ET is water limited and is mainly controlled by the surface state variables (i.e., soil moisture and NDVI). In contrast, at wet or densely planted conditions, ET is energy limited and is mainly influenced by atmospheric state variables (i.e., air temperature and specific humidity).

Finally, the long-term (1982–2011) total water storage anomaly (TWSA) is computed in the Mississippi River Basin (MRB) based on BTCH-integrated ET, using the *PER* method. The *PER* TWSA retrievals agree well with those of GRACE, and they both show an increasing trend over MRB from 1982 to 2011. Using the reconstructed long-term TWSA, the estimated TWS values increased by 0.06 (mm/year) over the MRB from 1982 to 2011.

**Author Contributions:** Methodology, Z.G., T.X., and X.H.; data processing and analyze, Z.G.; writing—original draft preparation, X.H.; writing—review and editing, X.H., T.X., and S.M.B., writing—review, editing, and supervision Y.X.; supervision, S.L., K.M., Y.Z., H.F., and J.Z.; project administration, T.X.; funding acquisition, T.X. All authors have read and agreed to the published version of the manuscript.

**Funding:** This work was funded by the National Natural Science Foundation of China (41531174 and 41671335).

**Acknowledgments:** This work was funded by the National Natural Science Foundation of China (41531174 and 41671335).

**Conflicts of Interest:** The authors declare no conflict of interest.

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
