# Peer review of "A Bayesian Three-Cornered Hat (BTCH) Method: Improving the Terrestrial Evapotranspiration Estimation"

_remotesensing, doi:10.3390/rs12050878_

Round 1

Reviewer 1 Report

This paper reports a study on the development of the Bayesian-based three-cornered hat (BTCH) method to improve ET estimation by integrating multiple ET products. The paper compares BTCH ET and several other ET products with reference ET datasets (eddy covariance and water-balance) and concludes that BTCH ET is the most accurate product. Additionally, the study analyzes the trends in ET over 30-year period and the relationships with precipitation, air temperature, soil moisture, and NDVI. The paper is generally well written, however, there are several issues that should be addressed.

Major issues:

The interpretation of the ET trends and the relationships with precipitation, air temperature, and NDVI are very simple and weak. More profound and detailed interpretations are suggested, especially for different RFC regions. Also, the results about the trends should be compared with previous studies from other publications. For the comparison between PER-retrieved TWSA and the GRACE TWSA (Section 4.4, statistical analyses (e.g., correlation, RMSD, etc.) should be used and presented. The whole section of discussion is actually a part of the analysis in the study, which should be placed in Methods and Results.

Minor issues:

L93. Change “Bayesian-based three-cornered hat (BTCH)” to BTCH.

L100-101. Barren is not a type of vegetation. Instead of barren land, shrubland should be included.

L137-139, “The 137 MODIS product provides monthly NDVI data with the spatial resolution of 1 km × 1 km 138 (https://ladsweb.modaps.eosdis.nasa.gov/tools-and-services/data-download-scripts/)”. The URL does link to a right website for MODIS data.

L150. Change “Bayesian-based three-cornered hat (BTCH)” to BTCH. Reference(s) is needed for BTCH.

L199. Ensemble mean (EM) method should be briefly described in here.

L217. Spell out PER method.

L222-224. Partial correlation analysis should be included in the method section.

L228-229. The spatial patterns of ET and vegetation cover are not consistent.

L238. Somewhere in this paragraph on in Figure 3, the number of samples should be specified.

L254. Need to indicate number of samples in this paragraph or in Figure 4.

L295. When the partial correlation is reported, the controlling variable should be indicated. For, example, partial correlation between A and B, controlling C and D.

L309. “1-meter” should be 12.5 km. (see Table 2)

Equation 1. “i’ and ‘t” should be defined.

Figure 1. Specify the landcover data source. The pink line should be explained in the caption.

Figure 3. Check the NDVI data source again. I doubt that the NDVI data were derived from MODIS, but AVHRR. However, if the data are from AVHRR, the resolution is 8 km, not 1 km.

Figure 8. Need a label for precipitation anomaly on the Y-axis. Although I guess precipitation anomaly shares the same Y-axis, it still needs a label.

Table 1. May add a column of elevation.

Table 2. The MODIS data records begin from late 1999.

Author Response

Response to Reviewer 1

Comments and Suggestions for Authors

This paper reports a study on the development of the Bayesian-based three-cornered hat (BTCH) method to improve ET estimation by integrating multiple ET products. The paper compares BTCH ET and several other ET products with reference ET datasets (eddy covariance and water-balance) and concludes that BTCH ET is the most accurate product. Additionally, the study analyzes the trends in ET over 30-year period and the relationships with precipitation, air temperature, soil moisture, and NDVI. The paper is generally well written, however, there are several issues that should be addressed.

Major Comments:

The interpretation of the ET trends and the relationships with precipitation, air temperature, and NDVI are very simple and weak. More profound and detailed interpretations are suggested, especially for different RFC regions. Also, the results about the trends should be compared with previous studies from other publications. For the comparison between PER-retrieved TWSA and the GRACE TWSA (Section 4.4, statistical analyses (e.g., correlation, RMSD, etc.) should be used and presented. The whole section of discussion is actually a part of the analysis in the study, which should be placed in Methods and Results.

To address the reviewer’s comments, the following sentence is added in our paper (Page 11-12, lines 314-324): “The results also show different trends of ET among the RFCs. Over Colorado and California-Nevada, the ET estimates show a small decrease over the 30-year period (1982–2011). While, there is a small increase in the ET estimates over Missouri, North central, Mid-Atlantic, Northeast, Lower Mississippi, and Ohio over the same period. Texas is mainly covered by the West Gulf and Arkansas basins, and experienced the most extreme one-year drought on record in 2011. The extreme one-year drought in 2011 has a large impact on the ET estimates [96,97]. Milly and Dunne (2001) [98] showed that increased ET in Mississippi has both climatological and anthropogenic dimensions. Walter et al. (2004) [99] studied ET across the CONUS based on precipitation and stream discharge data and showed that the ET has increased over the past 50 years. This study also reported an increasing trend in ET over the Mississippi basin and CONUS during 1982–2011, which is consistent with those of [98] and [99].”

For the comparison between PER-retrieved TWSA and the GRACE TWSA (Section 4.4, statistical analyses (e.g., correlation, RMSD, etc.) should be used and presented:

The following sentence is revised to our paper (Page 14, lines 372-374): “As shown in Figure 11 (bottom), the TWSA retrievals agree well with those of GRACE in terms of magnitude as well as seasonal and long-term variations. The RMSDs and R of TWSA estimates from PER method are 32.92 mm/month and 0.82.”

The whole section of discussion is actually a part of the analysis in the study, which should be placed in Methods and Results.

The Methods and Results sections are separate in our study just like other papers. There is a section called “Methodology” and an another section called “Results and Discussions”. There is no section called “Analysis”. The methodology section explains the Bayesian-based Three-Cornered Hat (BTCH) Method, the Water Balance Budget Method, and Long-term Total Water Storage Anomaly Reconstruction Approach. The “Results and Discussion” section includes analysis of ET product from the BTCH method, relationships between BTCH-integrated ET and climate factors, interannual variations of BTCH-integrated ET estimates, long-term reconstruction of GRACE total water storage anomaly, and time window of BTCH method.

Minor Comments:

L93. Change “Bayesian-based three-cornered hat (BTCH)” to BTCH.

Corrected.

L100-101. Barren is not a type of vegetation. Instead of barren land, shrubland should be included.

Corrected.

L137-139, “The 137 MODIS product provides monthly NDVI data with the spatial resolution of 1 km × 1 km 138 (https://ladsweb.modaps.eosdis.nasa.gov/tools-and-services/data-download-scripts/)”. The URL does link to a right website for MODIS data.

URL is corrected.

L150. Change “Bayesian-based three-cornered hat (BTCH)” to BTCH. Reference(s) is needed for BTCH.

Corrected. To address the reviewer’s comment, the references (Gray and Allan, 1974; Tavella and Premoli, 1994) are added to the BTCH method (Page 5, Line 155).

L199. Ensemble mean (EM) method should be briefly described in here.

The following sentence is added to our paper (Page 7, lines 205-207): “The ensemble mean (EM) method is also used to integrate the ten long-term gridded ET products, and generate an ET product. This method just simply calculates the average of ET products as follows,”

L217. Spell out PER method.

Done.

L222-224. Partial correlation analysis should be included in the method section.

Partial correlation analysis is added to our paper (Page 8, lines 239-250): “Partial correlation analysis [92,93] is used to obtain the correlation between collinear hydrologic variables (herein, Ta, P, and NDVI) that affect ET. The partial correlation analysis eliminates collinearity among the hydrologic variables (i.e., Ta, P, and NDVI) controlling ET. This analysis allows the impact of one or more variables (e.g., Ta and NDVI) to be eliminated when examining the relationship between another pair of variables (e.g., ET and P) [94,95]. Hence, the partial correlation coefficient quantifies the correlation between two variables (e.g., ET and P), controlling the other variables (i.e., Ta and NDVI).

     The partial correlation between the variables x and y, which are conditioned on the variable z can be calculated by,

,                            (23)

where Rxy,z is the partial correlation between x and y, given the control z. The variables Rxy, Rxz, Ryz are the correlation between x and y, x and z, and y and z, respectively.”

L228-229. The spatial patterns of ET and vegetation cover are not consistent.

The following sentence is tuned down in our paper (Page 8, Lines 254-255): “The spatial patterns of retrieved ET are fairly consistent with those of rainfall and to a lesser extent vegetation density.”

L238. Somewhere in this paragraph on in Figure 3, the number of samples should be specified.

The following sentence is added to the caption of Figure 4 (Page 10, Line 276): “There are 144, 336, and 300 samples for the DBF, ENF, and cropland, respectively.”

L254. Need to indicate number of samples in this paragraph or in Figure 4.

The following sentence is added to the caption of Figure 5 (Page 10, Line 285): “There are 360 samples over the CONUS.”

L295. When the partial correlation is reported, the controlling variable should be indicated. For, example, partial correlation between A and B, controlling C and D.

We added this sentences to our paper (Page 8, line 243-245) “Hence, the partial correlation coefficient quantifies the correlation between two variables (e.g., ET and P), controlling the other variables (i.e., Ta and NDVI).”

L309. “1-meter” should be 12.5 km. (see Table 2)

“1-meter” refers to depth. We removed 1-meter to avoid any confusion.

Equation 1. “i’ and ‘t” should be defined.

Done.

Figure 1. Specify the landcover data source. The pink line should be explained in the caption.

The following sentence is added to our paper (Page 3, Lines 102-103): “The land cover types are obtained from the MODIS land cover type product (MCD12Q1) (https://lpdaac.usgs.gov/products/mcd12q1v006/)”

The legend for the pink line is added to figure 1. It represents the Mississippi River Basin Boundary.

Figure 3. Check the NDVI data source again. I doubt that the NDVI data were derived from MODIS, but AVHRR. However, if the data are from AVHRR, the resolution is 8 km, not 1 km.

The following sentence is revised to our paper (Page 4, Lines 139-144): “Daily NDVI data are derived from the Advanced Very High Resolution Radiometer (AVHRR) instrument onboard the series of the National Oceanic and Atmospheric Administration (NOAA) satellites with the spatial resolution of 0.05° (https://climatedataguide.ucar.edu/climate-data/ndvi-normalized-difference-vegetation-index-noaa-avhrr).”

Figure 8. Need a label for precipitation anomaly on the Y-axis. Although I guess precipitation anomaly shares the same Y-axis, it still needs a label.

Corrected.

Table 1. May add a column of elevation.

Corrected.

Table 2. The MODIS data records begin from late 1999.

The NDVI data were derived from AVHRR (1981-2018).

References:

  1. Gray, J.E.; Allan, D.W. A method for estimating the frequency stability of an individual oscillator. Symposium on frequency control IEEE. 1974, 243–246.
  2. Tavella, P.; Premoli, A. Estimating the instabilities of N clocks by measuring differences of their readings. Metrologia. 1994, 30(5), 479–486. https://doi.org/10.1088/0026-1394/30/5/003
  3. Haan, C.T. Statistical methods in hydrology. Iowa State University Press, Ames, IA. 1977.
  4. de la Fuente, A.; Bing, N.; Hoeschele, I.; Mendes, P. Discovery of meaningful associations in genomic data using partial correlation coefficients. Bioinformatics. 2004, 20, 3565–3574. https://doi.org/10.1093/bioinformatics/bth445
  5. Thorndike, R.M. Correlational procedures for research. Gardner Press Inc., New York. 1976.
  6. Wilcox, B.P.; Wood, M.K.; Tromble, J.T.; Ward. T.J. A hand-portable single nozzle rainfall simulator designed for use on steep slopes. J. Range Manage. 1986, 39, 375–377
  7. Long, D.; Pan, Y.; Zhou, J.; Chen, Y.; Hou, X.Y.; Hong, Y.; et al. Global analysis of spatiotemporal variability in merged total water storage changes using multiple GRACE products and global hydrological models. Remote Sens. Environ. 2017, 192, 198–216. https://doi.org/10.1016/j.rse.2017.02.011
  8. Henn, B.; Painter, T.H.; Bormann, K.J.; McGurk, B.; Flint, A.L.; Flint, L.E.; et al. High-elevation evapotranspiration estimates during drought: using streamflow and NASA airborne snow observatory SWE observations to close the upper tuolumne river basin water balance. Water Resour. Res. 2018, 54, 746–766. https://doi.org/10.1002/2017WR020473
  9. Milly, P.C.D.; Dunne, K.A. Trends in evaporation and surface cooling in the Mississippi River basin. Geophys. Res. Lett. 2001, 28, 1219–1222. https://doi.org/10.1029/2000GL012321
  10. Walter, M.T.; Wilks, D.S.; Parlange, J.-Y.; Schneider, R.L. Increasing evapotranspiration from the conterminous United States. J. Hydrometeorol. 2004, 5, 405–408. https://doi.org/10.1175/1525-7541(2004)005<0405:IEFTCU>2.0.CO;2

Reviewer 2 Report

Line 63 - 65: Merging ET products with the multi-model ensembles method can generate an improve ET product with lower uncertainty [18,39]. Several studies showed that the ET estimates from the multi-model averaged method performs better than any individual model [4044].

It is not clear what is written. Multi-modal methods improves ET products with lower (?) uncertainty, while it performs better than individual models? I think this two sentences should rephrased.

Line 70: what is LE?

Line 93: it is not clear why authors used Bayesian methods. what are the problems with other methods?

Figure 6, represents air temperature, ET and NDVI. There is a positive relationship between ET and NDVI, where this Figure does not show this trend? what is the reason for this? Why ET, Ta and NDVI are all have the same data value range? This is also not clearly mentioned at Figure 7. Ialso would like to recommend to use a scatter plot showing relationship among ET, Ta, P and NDVI. That can give better understanding about changes and relations over time.

I would suggest the authors to make a better quality of figures with watermark texts.

There are other Remote Sensing methods for ET estimation based SEBAL algorithm and the SEBS model. I think it is required to compare the time-series results with the SEBAL and SEBS. There are other examples  of RS ET based methods which are not cited in this paper, like MODIS ET algorithm, ETWatch, and etc.

Author Response

Response to Reviewer 2

Comments and Suggestions for Authors

Comments:

Line 63 - 65: Merging ET products with the multi-model ensembles method can generate an improve ET product with lower uncertainty [18,39]. Several studies showed that the ET estimates from the multi-model averaged method performs better than any individual model [40–44]. It is not clear what is written. Multi-modal methods improves ET products with lower (?) uncertainty, while it performs better than individual models? I think this two sentences should rephrased.

To address the reviewer’s comments, the following sentences are modified as (Page 2, Lines 61-63): “Merging various ET products by the multi-model averaged method can generate an improved ET product with lower uncertainty [19,40]. Several studies showed that the ET estimates from the multi-model averaged method are better than those of individual models [41–45].”

The multi-model averaged method can provide a better ET product that has a lower uncertainty.

Line 70: what is LE?

We changed LE to ET.

Line 93: it is not clear why authors used Bayesian methods. what are the problems with other methods?

This is mentioned in our manuscript (Page 2, lines 71-74): “However, the results of weighted fusion method are affected by the accuracy of in situ observations and the spatial scale mismatch between flux towers ET observations (with 100-meter spatial representativeness) and gridded ET products (usually produced with 0.25° resolution) [48,49].”

Figure 6, represents air temperature, ET and NDVI. There is a positive relationship between ET and NDVI, where this Figure does not show this trend? what is the reason for this? Why ET, Ta and NDVI are all have the same data value range? This is also not clearly mentioned at Figure 7. I also would like to recommend to use a scatter plot showing relationship among ET, Ta, P and NDVI. That can give better understanding about changes and relations over time.

ET is affected by many variables (i.e., solar radiation, air temperature, specific humidity, land surface temperature, soil moisture, and NDVI) and not only NDVI. At dry and/or slightly vegetated areas, ET is water limited and is mainly controlled by the surface state variables (i.e., soil moisture and NDVI), and thus the correlation between ET and NDVI is most likely positive. In contrast, at wet and/or densely planted areas, ET is energy limited and is mainly influenced by atmospheric state variables (i.e., air temperature and specific humidity). Thus, the positive correlation between ET and NDVI vanishes. Because of the abovementioned factors, the scatter plots showing relationship between annual ET, and Ta, P, and NDVI over the CONUS from 1982-2011 would not be useful (please see below).

Fig. R1. Relationship between the annual ET, and Ta, P, and NDVI over the CONUS from 1982-2011.

To address the reviewer’s comment regarding the scatter plots, the following paragraph is added to our paper (page 12, lines 328-336): “Plots of 30-year-averaged ET against Ta, P, and NDVI over the 12 RFCs are shown in Figure 8. As shown, in general, ET grows by an increase in Ta, P, and NDVI over the 12 RFCs. ET estimates over Southeast, Lower Mississippi, and Ohio are higher than those of Colorado and California-Nevada because of higher vegetation density, precipitation, and air temperature. In West Gulf, low vegetation density and precipitation limit ET in spite of high air temperature. These results are consistent with those of [21] and [100].”

Figure 8. Plots of 30-year-averaged ET against Ta, P, and NDVI over the 12 RFCs. The red line represents the fitted linear regression.

The range of ET, Ta, NDVI, and P are not in our hands. They are obtained from different data sources for ET, Ta, NDVI, and P.

I would suggest the authors to make a better quality of figures with watermark texts.

All figures have a resolution of 350 dpi, and plotted by MATLAB.

There are other Remote Sensing methods for ET estimation based SEBAL algorithm and the SEBS model. I think it is required to compare the time-series results with the SEBAL and SEBS. There are other examples of RS ET based methods which are not cited in this paper, like MODIS ET algorithm, ETWatch, and etc.

Other remotely-sensed ET products (e.g., MOD16, SSEBop) are available over shorter time periods (i.e., less than 30 years that was used in this study). For example, MODIS and SSEBop ET products are available from 2000 to 2014. We found 10 long-term (30 years) ET products and we think they are enough to address the science questions of our paper.  

To address the reviewer’s comment, we added the following paragraph to our manuscript a number of RS ET methods in our paper (page 2, lines 84-86): “There are other ET products such as MOD16, ETWatch, and Simplified Surface Energy Balance (SSEBop) [33,56,57]. However, these ET products are not used in this study because they are available over shorter periods (i.e., less than 30 years).”

References:

  1. Zhang, K.; Kimball, J.S.; Running, S.W. A review of remote sensing based actual evapotranspiration estimation. Wiley Interdisciplinary Reviews: Water, 2016, 3(6), 834-853.
  2. Bateni, S.M.; Entekhabi, D.; Castelli, F. Mapping evaporation and estimation of surface control of evaporation using remotely sensed land surface temperature from a constellation of satellites. Water Resour. Res. 2013b, 49, 950–968. https://doi.org/10.1002/wrcr.20071
  3. Mu, Q.; Zhao, M.; Running, S.W. Improvements to a MODIS global terrestrial evapotranspiration algorithm. Remote Sens. Environ. 2011, 115(8), 1781–1800. http://dx. doi.org/10.1016/j.rse.2011.02.019.
  4. McCabe, M.F.; Ershadi, A.; Jimenez, C.; Miralles, D.G.; Michel, D.; Wood, E.F. The GEWEX LandFlux project: evaluation of model evaporation using tower-based and globally gridded forcing data. Geosci. Model Dev. 2016, 9, 283–305, https://doi.org/10.5194/gmd-9-283-2016
  5. Duan, Q.; Phillips, T.J. Bayesian estimation of local signal and noise in multimodel simulations of climate change. J. Geophys. Res. Atmos. 2010, 115, D18123. https://doi.org/10.1029/2009JD013654
  6. Houghton, J.T.A. Climate Change 2001: The Scientific Basis. Contribution of Working Group I to the Third Assessment Report of the Intergovernmental Panel on Climate Change, 2001, 892 pp., Cambridge Univ. Press, New York.
  7. Raftery, A.E.; Madigan, D.; Hoeting, J.A. Bayesian model averaging for linear regression models. J. Am. Stat. Assoc. 1997, 92(437), 179–191. https://doi.org/10.1080/01621459.1997.10473615
  8. Raftery, A.E.; Gneiting, T.; Balabdaoui, F.; Polakowski, M. Using Bayesian model averaging to calibrate forecast ensembles. Mon. Weather Rev. 2005, 133,1155–1174. https://doi.org/10.1175/MWR2906.1
  9. Wu, H.; Zhang, X.; Liang, S.; Yang, H.; Zhou, G. Estimation of clear-sky land surface longwave radiation from MODIS data products by merging multiple models. J. Geophys. Res. Atmos. 2012, 117, D22107. https://doi.org/10.1029/2012JD017567
  10. Hobeichi, S.; Abramowitz, G.; Evans, J.; Ukkola, A. Derived Optimal Linear Combination Evapotranspiration (DOLCE): a global gridded synthesis ET estimate. Hydrol. Earth Syst. Sci. 2018, 22(2), 1317.
  11. Burba, G.; Anderson, D. A Brief Practical Guide to Eddy Covariance Flux Measurements: Principles and Workflow Examples for Scientific and Industrial Applications, LI-COR Biosciences, Lincoln, Nebraska, USA. 2010.
  12. Wu, B.; Xiong, J.; Yan, N.; Yang, L.; Du, X. ETWatch for monitoring regional evapotranspiration with remote sensing. Adv. Rater Sci. 2008, 19(5), 671–678 (in Chinese with English abstract).
  13. Senay, G.B.; Bohms, S.; Singh, R.K.; Gowda, P.H.; Velpuri, N.M.; Alemu, H.; et al. Operational evapotranspiration mapping using remote sensing and weather datasets: a new parameterization for the SSEB approach. J. Am. Water Resour. Assoc. 2013, 49, 577–591. https://doi.org/10.1111/jawr.12057
  14. Bateni, S. M.; Entekhabi, D. Relative efficiency of land surface energy balance components. Water Resour. Res. 2012, 48, W04510. https://doi.org/10.1029/2011WR011357

Reviewer 3 Report

The submitted paper aims to develop a Bayesian-based three-cornered hat (BTCH) method in order improve terrestrial ET estimation by integrating multi-source ET products. Herein, authors proved the high performance of BTCH using several ET sources. Profound technical aspects of the study were implemented perfectly and explained sufficiently. Undoubtedly, authors invested huge amount of time and have made a great effort to produce this high-quality of research which is clearly structured; the language used largely appropriate. I would like to congratulate warmly the authors for producing this high-level research paper, and as final decision, I see that this manuscript in its form and level DESERVES TO BE ACCEPTED FOR PUBLICATION after considering the MINOR COMMENTS that I pointed below.

DETAILED COMMENTS:

The title is adequate for the content of the paper. The abstract is overloaded by huge detailed information, please try to rephrase it in a way to keep it simple & short and it covers all the sections of the paper from introduction to conclusion. I suggest for the authors to enrich the list of references by citing the following recent references in the introduction part:

Petropoulos, G. P., Ireland, G., Lamine, S., Griffiths, H. M., Ghilain, N., Anagnostopoulos, V., North M. R., Srivastava P. K. & Georgopoulou, H. Operational evapotranspiration estimates from SEVIRI in support of sustainable water management, International Journal of Applied Earth Observation and Geoinformation 49. 2016. 175–187 http://dx.doi.org/10.1016/j.jag.2016.02.006

Yao Y, Liang S, Li X, Chen J, Liu S, Jia K, Zhang X, Xiao Z, Fisher JB, Mu Q, Pan M, Liu M, Cheng J, Jiang B, Xie X, Grünwald T, Bernhofer C and Roupsard O (2017) Improving global terrestrial evapotranspiration estimation using support vector machine by integrating three process-based algorithms. Agricultural and Forest Meteorology 242:55-74. doi: https://doi.org/10.1016/j.agrformet.2017.04.011 

The sections of the paper are perfectly written and organised. All figures, tables and equations are good with high precision.                                                                The conclusion is well written and structured. Make sure to define ALL the acronyms form their first appearance in the paper. All references MUST BE CHECKED and formatted as required by MDPI-RS, also make sure that all the references have DOI number unless it is not available.

Author Response

Response to Reviewer 3

Comments and Suggestions for Authors

The submitted paper aims to develop a Bayesian-based three-cornered hat (BTCH) method in order improve terrestrial ET estimation by integrating multi-source ET products. Herein, authors proved the high performance of BTCH using several ET sources. Profound technical aspects of the study were implemented perfectly and explained sufficiently. Undoubtedly, authors invested huge amount of time and have made a great effort to produce this high-quality of research which is clearly structured; the language used largely appropriate. I would like to congratulate warmly the authors for producing this high-level research paper, and as final decision, I see that this manuscript in its form and level DESERVES TO BE ACCEPTED FOR PUBLICATION after considering the MINOR COMMENTS that I pointed below.

Comments:

The title is adequate for the content of the paper. The abstract is overloaded by huge detailed information, please try to rephrase it in a way to keep it simple & short and it covers all the sections of the paper from introduction to conclusion. I suggest for the authors to enrich the list of references by citing the following recent reference in the introduction part:

George P. Petropoulos; Gareth Ireland; Salim Lamine; Hywel M. Griffiths; Nicholas Ghilain; Vasilieios Anagnostopoulos; Matthew R. North; Prashant K. Srivastava; Hro Georgopoulou. Operational evapotranspiration estimates from SEVIRI in support of sustainable water management, International Journal of Applied Earth Observation and Geoinformation 49. 2016. 175–187 http://dx.doi.org/10.1016/j.jag.2016.02.006.

The abstract is revised and shortened.

We also referred to Petropoulos et al. (2016) in the introduction of our paper (page 1, lines 41-42): “The accurate estimation of ET is required for understanding the water resource management, water cycle, and climate change [3–6]”

The sections of the paper are perfectly written and organised. All figures, tables and equations are good with high precision. The conclusion is well written and structured. Make sure to define ALL the acronyms form their first appearance in the paper. All references MUST BE CHECKED and formatted as required by MDPI-RS, also make sure that all the references have DOI number unless it is not available.

Thanks for your kind help to improve our manuscript. All acronyms are defined in our paper. Also, all references are checked.

References:

  1. Er-Raki, S.; Chehbouni, A.; Boulet, G.; Williams, D. G. Using the dual approach of FAO-56 for partitioning ET into soil and plant components for olive orchards in a semi-arid region. Agric. Water Manage. 2010, 97, 1769-1778. https://doi.org/10.1016/j.agwat.2010.06.009
  2. Fisher, J.B.; Melton, F.; Middleton, E.; Hain, C.; Anderson, M.; Allen, R.; et al. The future of evapotranspiration: global requirements for ecosystem functioning, carbon and climate feedbacks, agricultural management, and water resources. Water Resour. Res. 2017, 53 (4), 2618–2626. https://doi.org/10.1002/2016WR020175
  3. Kustas, W.; Anderson, M. Advances in thermal infrared remote sensing for land surface modeling. Agric. For. Meteorol. 2009, 149, 2071–2081. https://doi.org/10.1016/j.agrformet.2009.05.016
  4. Petropoulos, G.P.; Ireland, G.; Lamine, S.; Griffiths, H.M.; Ghilain, G.; Anagnostopoulos, V.; et al. Operational evapotranspiration estimates from SEVIRI in support of sustainable water management. Int. J. Appl. Earth Obs. Geoinf. 2016, 49, 175–187. http://dx.doi.org/10.1016/j.jag.2016.02.006

Round 2

Reviewer 2 Report

I would like to appreciate authors effort on making better quality of visualized graphs.

Comment #1: Please make a reference to ETWatch framework in the introduction section.

Comment #2: I do not understand Figure 8! There should be a negative relationship between ET and temperature, whereas in Figure 8, there is a positive relationship between ET and temperature. What is the reason?

Comment #3: at line #423, Figure 14, how did authors calculate uncertainty and map it? it is only shortly mentioned at line #203, but not clear where the error variance comes with respect to what? is there any reference data? what is exactly ET(R)?

Comment #4: what is "<0405:IEFTCU>2.0.CO;2" at line #734? Please re-check all the references, as well.

Comment #5: I do not understand why authors only cited one group of authors for a periodic time-series analysis and cited several papers of only the same group of people at Reference numbers [88] to [90]. I think it is better to cite other references.

Author Response

Response to Reviewer 2

Comments and Suggestions for Authors

I would like to appreciate authors effort on making better quality of visualized graphs.

Thanks for your help to improve our manuscript. We have carefully revised the manuscript and responded your comments below.

Comments:

Please make a reference to ETWatch framework in the introduction section.

To address the reviewer’s comment, the references (Wu et al., 2010, 2016) are added to the introduction section.

The following sentence is revised in our paper (Page 2, lines 54-58): “Using the above-mentioned methods, various ET products (e.g., Gridded FLUXNET ET, GFET [10]; Moderate Resolution Imaging Spectroradiometer (MODIS) ET, MOD16 [34,35]; ETWatch [36,37]; Global Land Surface Satellite (GLASS) ET [38]; Global Land Evaporation and Amsterdam Model ET, GLEAM [39,40]; North American Land Data Assimilation System, NLDAS [31,41]) have been generated in last few decades.”

I do not understand Figure 8! There should be a negative relationship between ET and temperature, whereas in Figure 8, there is a positive relationship between ET and temperature. What is the reason?

Each point in Figure 8 represents one watershed. There are 12 RFCs and therefore there are 12 points in Fig. 8.

We respectfully disagree with the reviewer. Please see Bateni and Entekhabi (2012) (Relative efficiency of land surface energy balance components: WRR, volume 48). They theoretically proved that ET increases with Ta, depending on the soil moisture availability. For high soil moisture values, ET increases rapidly with Ta. For low soil moisture values, ET increases slightly with Ta. They also verified their theoretical derivations/results by field data. The physical reason for the increase of ET with air temperature is explained comprehensively in Bateni and Entekhabi (2012).   

at line #423, Figure 14, how did authors calculate uncertainty and map it? it is only shortly mentioned at line #203, but not clear where the error variance comes with respect to what? is there any reference data? what is exactly ET(R)?

The following sentences are added to our paper:

(Page 6, lines 180-181): “Since ETt is not available, the difference between (N-1) ET products and a reference ET product (ETR) (chosen arbitrarily from N ET products) can be expressed as, ….”

(Page 6, lines 191-193): “and matrix R is defined as,

,                         (12)

where σij = cov(É›i, É›j).”

(Page 7, lines 204-207): “The square root of the diagonal elements of R (i.e., σ11, σ22, …, σNN) represent the relative uncertainty of each ET product [43]. The readers are referred to Long et al. (2014) [42] and Xu et al. (2019) [43] for detailed information on the TCH approach.”

what is "<0405:IEFTCU>2.0.CO;2" at line #734? Please re-check all the references, as well.

Thank you. The DOI address is correct. We rechecked all the references in our manuscript.

I do not understand why authors only cited one group of authors for a periodic time-series analysis and cited several papers of only the same group of people at Reference numbers [88] to [90]. I think it is better to cite other references.

We have provided references from different groups including Rahman et al. (2017) [92] and Fathian et al. (2016) [93].

References:

  1. Jung, M.; Reichstein, M.; Ciais, P.; Seneviratne, S.I.; Sheffield, J.; Goulden, M. L.; et al. Recent decline in the global land evapotranspiration trend due to limited moisture supply. Nature, 2010, 467(7318), 951–954. https://doi.org/10.1038/nature09396.
  2. Bateni, S.M.; Entekhabi, D.; Jeng, D.S. Variational assimilation of land surface temperature and the estimation of surface energy balance components. J. Hydrol. 2013, 481, 143–156. https://doi.org/10.1016/j.jhydrol.2012.12.039
  3. Xia, Y.; Hao, Z.; Shi, C.; Li, Y.; Meng, J.; Xu, T.; et al. Regional and Global Land Data Assimilation Systems: Innovations, Challenges, and Prospects, J. Meteorol. Res. 2019, 33(2), 159–189.
  4. Mu, Q.; Heinsch, F.A.; Zhao, M.; Running, S.W. Development of a global evapotranspiration algorithm based on MODIS and global meteorology data. Remote Sens. Environ. 2007, 111, 519–536. http://dx.doi.org/10.1016/j.rse.2006.07.007.
  5. Mu, Q.; Zhao, M.; Running, S.W. Improvements to a MODIS global terrestrial evapotranspiration algorithm. Remote Sens. Environ. 2011, 115(8), 1781–1800. http://dx. doi.org/10.1016/j.rse.2011.02.019.
  6. Wu, B.; Xiong, J.; Yan, N. Etwatch: Models and methods. J. Remote Sens. 2010, 15, 224–230.
  7. Wu, B.; Zhu, W.; Yan, N.; Feng, X.; Xing, Q.; Zhuang, Q. An improved method for deriving daily evapotranspiration estimates from satellite estimates on cloud-free days. IEEE J. Sel. Top. Appl. Earth Obs. Remote Sens. 2016, 9, 1323–1330.
  8. Yao, Y.; Liang, S.; Li, X.; Hong, Y.; Fisher, J.B.; Zhang, N.; et al. Bayesian multimodel estimation of global terrestrial latent heat flux from eddy covariance, meteorological, and satellite observations. J. Geophys. Res. Atmos. 2014, 119, 4521–4545.
  9. Martens, B.; Miralles, D.; Lievens, H.; Fernández-Prieto, D.; Verhoest, N. E. Improving terrestrial evaporation estimates over continental Australia through assimilation of SMOS soil moisture. Int. J. Appl. Earth Obs. Geoinf. 2016, 48, 146–162.
  10. Martens, B.; Miralles, D.G.; Lievens, H.; van der Schalie, R.; de Jeu, R.A.M.; Fernández–Prieto, D.; et al. GLEAM v3: satellite–based land evaporation and root–zone soil moisture. Geosci. Model Dev. 2017, 10(5), 1903–1925.
  11. Mitchell, K.E.; Lohmann, D.; Houser, P.R.; Wood, E.F.; Schaake, J.C.; Robock, A.; et al. The multi–institution North American Land Data Assimilation System (NLDAS): utilizing multiple GCIP products and partners in a continental distributed hydrological modeling system. J. Geophys. Res. 2004, 109, D07S90, DOI:10.1029/2003JD003823.
  12. Long, D.; Longuevergne, L.; Scanlon, B.R. 2014. Uncertainty in evapotranspiration from land surface modeling, remote sensing, and GRACE satellites. Water Resour. Res. 2014, 50(2), 1131–1151. https://doi.org/10.1002/2013WR014581
  13. Xu, T.R.; Guo, Z.X.; Xia, Y.L.; Ferreira, V.G.; Liu, S.M.; Wang, K.C.; et al. Evaluation of twelve evapotranspiration products from machine learning, remote sensing and land surface models over conterminous united states. J. Hydrol. 2019, 578, 124105. https://doi.org/10.1016/j.jhydrol. 2019.124105
  14. Rahman, M.; Yunsheng, L.; Sultana, N. Analysis and prediction of rainfall trends over Bangladesh using Mann–Kendall, Spearman’s rho tests and ARIMA model. Meteorol. Atmos. Phys. 2017, 129, 409–424.
  15. Fathian, F.; Dehghan, Z.; Bazrkar, M.H.; Eslamian, S. Trends in hydrological and climatic variables affected by four variations of the Mann-Kendall approach in Urmia Lake basin. Iran. Hydrol. Sci. J. 2016, 61(5), 892–904.
